# Multivariate Probabilistic Time Series Forecasting with Correlated Errors

**Vincent Zhihao Zheng**
McGill University
Montréal, QC, Canada
zhihao.zheng@mail.mcgill.ca

**Lijun Sun**[*]
McGill University
Montréal, QC, Canada
lijun.sun@mcgill.ca

## Abstract

Accurately modeling the correlation structure of errors is critical for reliable uncertainty quantification in probabilistic time series forecasting. While recent deep learning models for multivariate time series have developed efficient parameterizations for time-varying contemporaneous covariance, but they often assume temporal independence of errors for simplicity. However, real-world data often exhibit significant error autocorrelation and cross-lag correlation due to factors such as missing covariates. In this paper, we introduce a plug-and-play method that learns the covariance structure of errors over multiple steps for autoregressive models with Gaussian-distributed errors. To ensure scalable inference and computational efficiency, we model the contemporaneous covariance using a low-rank-plus-diagonal parameterization and capture cross-covariance through a group of independent latent temporal processes. The learned covariance matrix is then used to calibrate predictions based on observed residuals. We evaluate our method on probabilistic models built on RNNs and Transformer architectures, and the results confirm the effectiveness of our approach in improving predictive accuracy and uncertainty quantification without significantly increasing the parameter size.

## 1 Introduction

Uncertainty quantification is crucial in time series forecasting, especially for applications that need more detailed insights than point forecasts. Probabilistic time series forecasting with deep learning (DL) has attracted attention for its ability to capture complex, nonlinear dependencies and provide the probability distribution of target variables [1, 2]. In multivariate time series, autoregressive models are widely used for probabilistic forecasting [3–5], modeling the joint one-step-ahead predictive distribution and generating multistep-ahead predictions in a rolling manner. To enable scalable learning, these models often assume that errors are independent over time. Typically, time series variables follow a Gaussian distribution $\mathbf{z}_t = m(\mathbf{h}_t) + \boldsymbol{\eta}_t$, where $m(\cdot)$ is the mean function and $\boldsymbol{\eta}_t \sim \mathcal{N}(\mathbf{0}, \boldsymbol{\Sigma}_t)$ is a stochastic error process with contemporaneous covariance matrix $\boldsymbol{\Sigma}_t$. The assumption of time-independence implies $\mathrm{Cov}(\boldsymbol{\eta}_s, \boldsymbol{\eta}_t) = \mathbf{0}$, $\forall s \neq t$. This holds when the model can account for all correlations between successive time steps through hidden states determined by previous values. However, real-world data often violate this assumption, as residuals exhibit substantial cross-correlation due to omission of important covariates and model misspecification.

Modeling error autocorrelation (or cross-correlation) is a key area of research in statistical time series models. A common approach is to assume that the error series follows a dependent temporal process, such as an autoregressive integrated moving average (ARIMA) model [6]. Deep learning models face similar challenges. Previous studies have attempted to incorporate temporally-correlated errors into the training process by modifying the loss function [7, 8]. However, these methods, based on

---

[*]Corresponding author.

38th Conference on Neural Information Processing Systems (NeurIPS 2024).

deterministic output, are not easily applicable to probabilistic forecasting models, particularly in a multivariate setting. A notable innovation is the batch training method introduced by Zheng et al. [9], which trains a univariate probabilistic forecasting model using generalized least squares (GLS) loss over batched errors. This approach parameterizes a dynamic covariance matrix to capture error autocorrelation, which is then used to calibrate the predictive distribution of time series variables. While this method consistently improves probabilistic forecasting performance compared to naive training (i.e., without considering autocorrelated errors); however, it is only applicable to univariate models, such as DeepAR [10].

In this paper, we introduce an efficient method for learning error cross-correlation in multivariate probabilistic forecasting models. Our focus is on deep learning models that are autoregressive with Gaussian-distributed errors. Modeling cross-correlation in multivariate models presents challenges due to increased dimensionality, as the covariance matrix scales with the number of time series $N$. To address this computational challenge, we propose characterizing error cross-correlation through a set of independent latent temporal processes using a low-rank parameterization of the covariance matrix. This approach prevents the computational cost from growing with the number of time series. Our method offers a general-purpose approach to multivariate probabilistic forecasting models, offering significantly improved predictive accuracy.

**Contributions:**

1. We introduce a plug-and-play method for training autoregressive multivariate probabilistic forecasting models using a redesigned GLS loss. (§4)
2. We propose an efficient parameterization of the error covariance matrix across multiple steps, enabling efficient computation of its inverse and determinant through matrix inversion and determinant lemmas. (§4.1)
3. The learned covariance matrix is used to fine-tune the predictive distribution based on observed residuals. (§4.2)
4. We demonstrate that the proposed method effectively captures error cross-correlation and improves prediction quality. Notably, these improvements are achieved through a statistical formulation without significantly increasing the size of model parameters. (§5)

## 2 Probabilistic Time Series Forecasting

Denote $\mathbf{z}_t = [z_{1,t}, \ldots, z_{N,t}]^\top \in \mathbb{R}^N$ as the vector of time series variables at time step $t$, where $N$ is the number of time series. Probabilistic time series forecasting can be formulated as estimating the joint conditional distribution $p\left(\mathbf{z}_{T+1:T+Q} \mid \mathbf{z}_{T-P+1:T}; \mathbf{x}_{T-P+1:T+Q}\right)$ given the observed history $\{\mathbf{z}_t\}_{t=1}^T$, where $\mathbf{z}_{t_1:t_2} = [\mathbf{z}_{t_1}, \ldots, \mathbf{z}_{t_2}]$ and $\mathbf{x}_t$ are known time-dependent covariates (e.g., time of day, day of week) for all future time steps. In essence, the problem involves predicting the time series values for $Q$ future time steps using all available covariates and $P$ steps of historical time series data:

$$p\left(\mathbf{z}_{T+1:T+Q} \mid \mathbf{z}_{T-P+1:T}; \mathbf{x}_{T-P+1:T+Q}\right) = \prod_{t=T+1}^{T+Q} p\left(\mathbf{z}_t \mid \mathbf{z}_{t-P:t-1}; \mathbf{x}_{t-P:t}\right), \tag{1}$$

which becomes an autoregressive model that can be used for either one-step-ahead ($Q = 1$) or multistep-ahead forecasting in a rolling manner. When performing multistep-ahead forecasting, samples are drawn in the prediction range ($t \geq T + 1$) and fed back for the next time step until the end of the desired prediction range. In neural networks, the conditioning information is commonly encoded into a state vector $\mathbf{h}_t$. Hence, Eq. (1) can be expressed more concisely:

$$p\left(\mathbf{z}_{T+1:T+Q} \mid \mathbf{z}_{T-P+1:T}; \mathbf{x}_{T-P+1:T+Q}\right) = \prod_{t=T+1}^{T+Q} p\left(\mathbf{z}_t \mid \mathbf{h}_t\right), \tag{2}$$

where $\mathbf{h}_t$ is mapped to the parameters of a parametric distribution (e.g., multivariate Gaussian).

Existing autoregressive models typically assume that the error at each time step is independent, meaning that $\mathbf{z}_t$ follows a multivariate Gaussian distribution:

$$\mathbf{z}_t \mid \mathbf{h}_t \sim \mathcal{N}\left(\boldsymbol{\mu}(\mathbf{h}_t), \boldsymbol{\Sigma}(\mathbf{h}_t)\right), \tag{3}$$

where $\boldsymbol{\mu}(\cdot)$ and $\boldsymbol{\Sigma}(\cdot)$ map $\mathbf{h}_t$ to the mean and covariance parameters of a multivariate Gaussian distribution. This formulation can be decomposed as $\mathbf{z}_t = \boldsymbol{\mu}_t + \boldsymbol{\eta}_t$ with $\boldsymbol{\eta}_t \sim \mathcal{N}(\mathbf{0}, \boldsymbol{\Sigma}_t)$. The temporally independent error assumption corresponds to $\mathrm{Cov}(\boldsymbol{\eta}_s, \boldsymbol{\eta}_t) = \mathbf{0}$ for any time points $s$ and

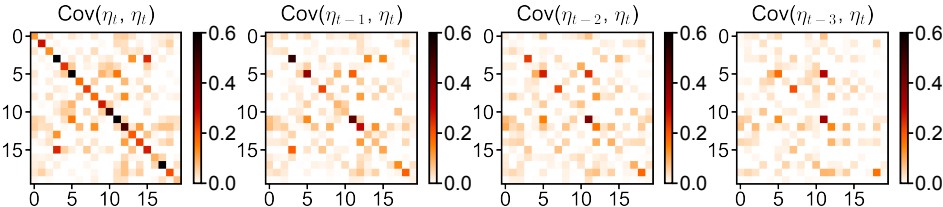

Figure 1: Contemporaneous covariance matrix $\text{Cov}(\boldsymbol{\eta}_t, \boldsymbol{\eta}_t)$ and cross-covariance matrix $\text{Cov}(\boldsymbol{\eta}_{t-\Delta}, \boldsymbol{\eta}_t), \Delta = 1, 2, 3$, calculated based on the one-step-ahead prediction residuals of GP-Var on a batch of time series from the m4_hourly dataset. For visualization clarity, covariance are clipped to the range $[0, 0.6]$.

$t$ where $s \neq t$. Fig. 1 provides an empirical example of the contemporaneous covariance matrix $\text{Cov}(\boldsymbol{\eta}_t, \boldsymbol{\eta}_t)$ and cross-covariance matrix $\text{Cov}(\boldsymbol{\eta}_{t-\Delta}, \boldsymbol{\eta}_t), \Delta = 1, 2, 3$. The results are calculated based on the prediction residuals of GPVar [3] on the m4_hourly dataset. While multivariate models primarily focus on contemporaneous covariance, the residuals clearly exhibit temporal dependence, as $\text{Cov}(\boldsymbol{\eta}_{t-\Delta}, \boldsymbol{\eta}_t) \neq \mathbf{0}$. This non-zero cross-covariance suggests that residuals still contain valuable information, which can be leveraged to improve predictions.

## 3 Related Work

### 3.1 Probabilistic Time Series Forecasting

Probabilistic forecasting aims to model the probability distribution of target variables, unlike deterministic forecasting, which produces only point estimates. There are two main approaches: parametric probability density functions (PDFs) and quantile functions [2]. For example, MQ-RNN [11] generates quantile forecasts using a sequence-to-sequence (Seq2Seq) RNN architecture. In contrast, PDF-based approaches assume a specific distribution (e.g., Gaussian, Poisson) and use neural networks to generate the distribution parameters. DeepAR [10], for instance, uses an RNN to model hidden state transitions, while its multivariate version, GPVar [3], employs a Gaussian copula to transform observations into Gaussian variables, assuming a joint multivariate Gaussian distribution.

Neural networks can also generate probabilistic model parameters. The deep state space model (SSM) [12] uses an RNN to generate SSM parameters. The normalizing Kalman filter (NKF) [13] combines normalizing flows (NFs) with the linear Gaussian state space model (LGM) to model nonlinear dynamics and evaluate the PDF of observations. NKF uses RNNs to produce LGM parameters at each time step, then transforms the LGM output into observations using NFs. Wang et al. [14] proposed the deep factor model, which includes a deterministic global component parameterized by an RNN and a random component from any classical probabilistic model (e.g., Gaussian white noise) to represent random effects. Some methods improve expressive conditioning for probabilistic forecasting by using Transformer instead of RNNs to model latent state dynamics, thus breaking the Markovian assumption in RNNs [15]. Other approaches adopt more flexible distribution forms, including normalizing flows [4], diffusion models [5], and copulas [16, 17]. For a recent and comprehensive review, we refer readers to Benidis et al. [2].

### 3.2 Modeling Correlated Errors

Error correlation in time series has been extensively studied in econometrics and statistics [18, 6, 19]. In multivariate time series, correlation structure is characterized by contemporaneous covariance $\text{Var}(\boldsymbol{\eta}_t) = \text{Cov}(\boldsymbol{\eta}_t, \boldsymbol{\eta}_t)$ and cross-covariance $\text{Cov}(\boldsymbol{\eta}_{t-\Delta}, \boldsymbol{\eta}_t)$. Cross-covariance includes both the autocovariance of errors $\text{Cov}(\eta_{i,t-\Delta}, \eta_{i,t})$ and the cross-lag covariance $\text{Cov}(\eta_{i,t-\Delta}, \eta_{j,t})$ between pairs of components in the multivariate series. Contemporaneous covariance captures the correlation among individual time series at a specific point in time. In the univariate setting, DeepAR [10] achieves probabilistic forecasting by modeling the contemporaneous covariance, assuming that errors are independent over time. To address autocorrelation, Sun et al. [7] re-parameterized the input and output of neural networks to model first-order error autocorrelation, effectively capturing serially

correlated errors using an AR(1) process. This method improves the performance of one-step-ahead neural forecasting models, allowing joint optimization of base and error regressors, but is limited to deterministic models. In spatial modeling, Saha et al. [20] introduced the RF-GLS model, which uses random forests to estimate nonlinear covariate effects and Gaussian processes (GP) to model spatial random effects. The RF-GLS model assumes that the error process follows an AR($p$) process to accommodate autocorrelated errors. Zheng et al. [9] proposed training a probabilistic forecasting model with a GLS loss that explicitly models the time-varying autocorrelation of batched error terms, extending DeepAR to incorporate autocorrelated errors.

In the multivariate setting, most existing work focuses on modeling contemporaneous covariance, assuming that $\boldsymbol{\eta}_t$ is independently distributed, which implies $\mathrm{Cov}(\boldsymbol{\eta}_{t-\Delta}, \boldsymbol{\eta}_t) = \mathbf{0}$. For example, GPVar [3] generalizes DeepAR [10] to account for correlations between time series by viewing the distribution of time series variables as a Gaussian process. In Seq2Seq models, correlations can span across series and forecasting steps, as predictions for future time steps are generated simultaneously. Since predictions for future time steps are generated simultaneously, we refer to these correlations as contemporaneous correlations within the scope of this study. Choi et al. [21] introduced a dynamic mixture of matrix Gaussian distributions to capture contemporaneous covariance of errors in Seq2Seq models. One exception that explicitly models error cross-correlation is [8], where the authors assume that the matrix-variate error term of a multivariate Seq2Seq model follows a matrix autoregressive (AR) process with seasonal lags. However, applying this technique to probabilistic forecasting models is not straightforward.

To the best of our knowledge, our work is the first to model cross-covariance in multivariate probabilistic time series forecasting. The closest related studies are by Zheng et al. [9] and Zheng et al. [8]. Zheng et al. [9] applies GLS loss in the temporal domain to model autocorrelated errors, but their approach is tailored for univariate time series. Zheng et al. [8] models cross-covariance in multivariate forecasting models, but their method is limited to deterministic models and requires predefined seasonal lags in the error autoregressive process. Our work extends [9] to the multivariate setting, enabling the modeling of the correlation structure of multivariate errors across multiple steps. In addition, we distinguish our approach from methods that directly model the distribution of time series variables, such as Copulas [16, 17], where no decomposition of the error term is provided.

## 4 Our Method

Our methodology builds upon the formulation outlined in Eq. (2), employing an autoregressive model as its foundational framework. Using an RNN as an example, a probabilistic forecasting model consists of two components. Firstly, it incorporates a transition model $f_\Theta$ to capture the dynamics of state transitions $\mathbf{h}_t = f_\Theta(\mathbf{h}_{t-1}, \mathbf{z}_{t-1}, \mathbf{x}_t)$, thus inherently having autoregressive properties. Second, it integrates a distribution head, represented by $\theta$, which maps $\mathbf{h}_t$ to the parameters of the desired probability distribution. Following GPVar [3], our approach employs the multivariate Gaussian distribution as the distribution head. The time series variable can be decomposed into a deterministic mean component and a random error component $\mathbf{z}_t = \boldsymbol{\mu}_t + \boldsymbol{\eta}_t$, where $\boldsymbol{\eta}_t \sim \mathcal{N}(\mathbf{0}, \boldsymbol{\Sigma}_t)$. To efficiently model the covariance $\boldsymbol{\Sigma}_t$ for large $N$, GPVar adopts a low-rank-plus-diagonal parameterization $\boldsymbol{\Sigma}_t = \boldsymbol{L}_t \boldsymbol{L}_t^\top + \mathrm{diag}(\mathbf{d}_t)$, where $\boldsymbol{L}_t \in \mathbb{R}^{N \times R}$ ($R \ll N$) and $\boldsymbol{d}_t \in \mathbb{R}_+^N$. Autoregressive models based on Gaussian likelihood typically assume that $\boldsymbol{\eta}_t$ are independently distributed following a multivariate Gaussian distribution. The log-likelihood of the distribution serves as the loss function for optimizing the model:

$$\mathcal{L} = \sum\nolimits_{t=1}^{T} \log p(\mathbf{z}_t \mid \theta(\mathbf{h}_t)) \propto \sum\nolimits_{t=1}^{T} -\frac{1}{2}[\ln|\boldsymbol{\Sigma}_t| + \boldsymbol{\eta}_t^\top \boldsymbol{\Sigma}_t^{-1} \boldsymbol{\eta}_t]. \tag{4}$$

The parameters $\theta(\mathbf{h}_t)$ are parameterized as $(\boldsymbol{\mu}_t, \boldsymbol{L}_t, \mathbf{d}_t)$, where $\boldsymbol{\mu}_t \in \mathbb{R}^N$ represents the mean vector of the distribution. $\boldsymbol{L}_t$ and $\mathbf{d}_t$ correspond to the covariance factor and diagonal elements in the low-rank parameterization of the multivariate Gaussian distribution. We use shared mapping functions for all time series:

$$\begin{aligned}
\mu_i(\mathbf{h}_{i,t}) &= \tilde{\mu}(\mathbf{h}_{i,t}) = \mathbf{w}_\mu^\top \mathbf{h}_{i,t}, \\
d_i(\mathbf{h}_{i,t}) &= \tilde{d}(\mathbf{h}_{i,t}) = \log(1 + \exp(\mathbf{w}_d^\top \mathbf{h}_{i,t})), \\
l_i(\mathbf{h}_{i,t}) &= \tilde{l}(\mathbf{h}_{i,t}) = W_l \mathbf{h}_{i,t},
\end{aligned} \tag{5}$$

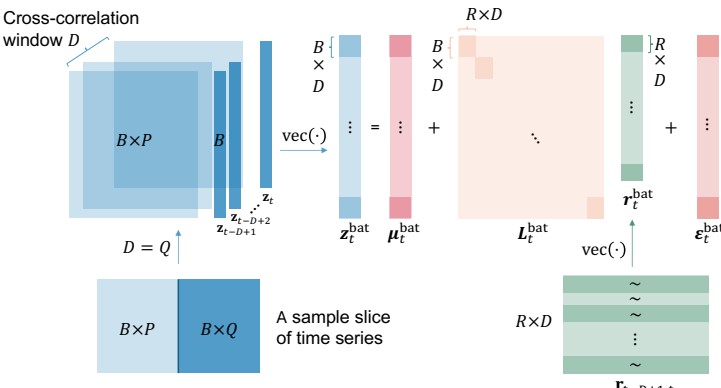

Figure 2: Graphic illustration of Eq. (8), where $B$ is the number of time series in a batch, $R$ is the rank of the covariance factor, $D$ is the time window we consider cross-correlation, $P$ and $Q$ are the conditioning range and prediction range. Cross-correlation is modeled by introducing correlation in each row of matrix $\mathbf{r}_{t-D+1:t}$.

where $\mathbf{h}_{i,t} \in \mathbb{R}^H$, $\mathbf{w}_\mu \in \mathbb{R}^H$, $\mathbf{w}_d \in \mathbb{R}^H$, and $W_l \in \mathbb{R}^{R \times H}$ are parameters. Since the parameters of the mapping functions are shared across all time series, we can use a random subset of time series to compute the Gaussian likelihood-based loss in each optimization step, as any subset of $\mathbf{z}_t$ will still follow a multivariate Gaussian distribution. In other words, we can train the model with a substantially reduced batch size $B < N$.

## 4.1 Training with Correlated Errors

We build upon the approach introduced in [9] to address cross-correlated errors in a multivariate context by introducing time-dependent error terms $\boldsymbol{\eta}_t$ into the GLS loss. In many existing deep probabilistic forecasting models, such as GPVar [3], a training batch typically consists of a sample slice of $B$ time series spanning a temporal length of $P + Q$, where $P$ is the conditioning range and $Q$ is the prediction range. The Gaussian likelihood is evaluated independently at each time step within the prediction range through one-step-ahead predictions. However, this approach overlooks the serial correlation of errors across consecutive time steps. To address this limitation, we propose modifying the likelihood function by introducing a dynamic covariance that accommodates the temporal dependence of the error term, as illustrated in Fig. 2. To achieve this, we organize $D$ smaller slices of time series with a temporal length of $P + 1$ (i.e., $Q = 1$), sorted by the prediction start time in sequential order, where $D$ represents the time horizon over which we consider cross-correlation. The new batch structure effectively reconstructs the conventional training batch, covering the same time horizon when $D = Q$. An example of the collection of target time series variables in a batch covering cross-correlation horizon $D$ is given by

$$
\begin{aligned}
\mathbf{z}_{t-D+1} &= \boldsymbol{\mu}_{t-D+1} + \boldsymbol{\eta}_{t-D+1}, \\
\mathbf{z}_{t-D+2} &= \boldsymbol{\mu}_{t-D+2} + \boldsymbol{\eta}_{t-D+2}, \\
&\cdots \\
\mathbf{z}_t &= \boldsymbol{\mu}_t + \boldsymbol{\eta}_t,
\end{aligned}
\tag{6}
$$

where for time point $t'$, $\boldsymbol{\mu}_{t'}$, $\boldsymbol{L}_{t'}$ and $\mathbf{d}_{t'}$ are the outputs of the model. The covariance parameterization in GPVar corresponds to

$$
\boldsymbol{\eta}_{t'} = \boldsymbol{L}_{t'}\mathbf{r}_{t'} + \boldsymbol{\varepsilon}_{t'},
\tag{7}
$$

where $\mathbf{r}_{t'} \sim \mathcal{N}(\mathbf{0}, \mathbf{I}_R)$ is a low-dimensional latent variable, and $\boldsymbol{\varepsilon}_{t'} \sim \mathcal{N}(\mathbf{0}, \operatorname{diag}(\mathbf{d}_{t'}))$ is an additional error independent of $\mathbf{r}_{t'}$. We denote $\mathbf{z}_t^{\text{bat}} = \operatorname{vec}(\mathbf{z}_{t-D+1:t}) \in \mathbb{R}^{DB}$ as the collection of target time series variables in a batch, where $\operatorname{vec}(\cdot)$ is an operator that stacks all the columns of a matrix into a vector. Similarly, we define $\boldsymbol{\mu}_t^{\text{bat}} \in \mathbb{R}^{DB}$, $\mathbf{r}_t^{\text{bat}} \in \mathbb{R}^{DR}$, $\boldsymbol{\varepsilon}_t^{\text{bat}} \in \mathbb{R}^{DB}$, $\mathbf{d}_t^{\text{bat}} \in \mathbb{R}_+^{DB}$, and $\boldsymbol{L}_t^{\text{bat}} = \operatorname{blkdiag}(\{\boldsymbol{L}_{t'}\}_{t'=t-D+1}^t) \in \mathbb{R}^{DB \times DR}$, where $\boldsymbol{L}_t^{\text{bat}}$ has a block diagonal structure (see Fig. 2). The batch-wise decomposition is then expressed as

$$
\mathbf{z}_t^{\text{bat}} = \boldsymbol{\mu}_t^{\text{bat}} + \boldsymbol{L}_t^{\text{bat}}\mathbf{r}_t^{\text{bat}} + \boldsymbol{\varepsilon}_t^{\text{bat}}.
\tag{8}
$$

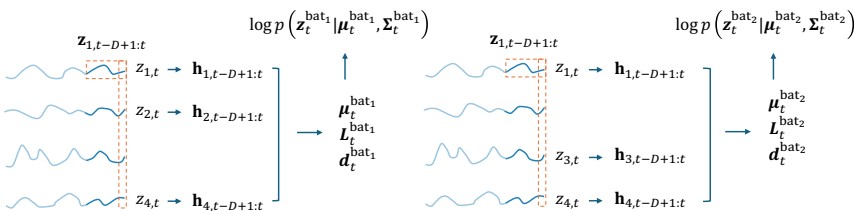

Figure 3: Illustration of the training process. Following [3], time series dimensions are randomly sampled, and the base model (e.g., RNNs) is unrolled for each dimension individually (e.g., 1, 2, 4, followed by 1, 3, 4 as depicted). The model parameters are shared across all time series dimensions. A batch of time series variables $z_t^{\text{bat}}$ contains time series vectors $z_t$ covering time steps from $t - D + 1$ to $t$. In contrast to [3], our approach explicitly models dependencies over the extended temporal window from $t - D + 1$ to $t$ during training.

The default GPVar model assumes the latent variable $r_t$ is temporally independent, meaning $\text{Cov}\left(r_s, r_t\right) = 0, \forall s \neq t$. However, this assumption cannot capture the potential cross-correlation in the errors. To address this, we introduce temporal dependencies in the latent variable within a batch by assuming $r_t^{\text{bat}} \sim \mathcal{N}\left(0, C_t \otimes I_R\right)$, where $C_t$ is a dynamic $D \times D$ correlation matrix. This approach assumes that the rows in the matrix $\mathbf{r}_{t-D+1:t} = [r_{t-D+1}, \ldots, r_t]$ are independent and identically distributed, following $\mathcal{N}\left(0, C_t\right)$. To efficiently capture dynamic patterns over time, we follow Zheng et al. [9] and express $C_t$ as a dynamic weighted sum base kernel matrices: $C_t = \sum_{m=1}^{M} w_{m,t} K_m$, where $w_{m,t} \geq 0$ (with $\sum_m w_{m,t} = 1$) represents the weights for each component. For simplicity, we model each component $K_m$ using a kernel matrix generated from a squared-exponential (SE) kernel function, where the $(i, j)$-th entry is $K_m^{ij} = \exp(-\frac{(i-j)^2}{l_m^2})$, with different lengthscales $l_m$ (e.g., $l = 1, 2, 3, \ldots$). In addition, we incorporate an identity matrix into the additive structure to account for the independent noise process. This parameterization ensures that $C_t$ is a positive definite symmetric matrix with unit diagonals, making it a valid correlation matrix. The weights for these components are derived from the hidden state $\mathbf{h}_t$ at each time step through a small neural network, with the number of nodes in the output layer set to $M$ (i.e., the number of components). A softmax layer is used to ensure that these weights are summed up to 1. Note that the parameters of this network will be learned simultaneously with those of the base model.

Marginalizing out $r_t^{\text{bat}}$ in Eq. (8), we have $z_t^{\text{bat}} \sim \mathcal{N}\left(\mu_t^{\text{bat}}, \Sigma_t^{\text{bat}}\right)$ with covariance

$$\Sigma_t^{\text{bat}} = (L_t^{\text{bat}})(C_t \otimes I_R)(L_t^{\text{bat}})^{\top} + \text{diag}(d_t^{\text{bat}}). \tag{9}$$

It is straightforward to derive that for any $i, j \in \{0, 1, \ldots, D-1\}$ and $i \neq j$, the proposed model creates cross-covariance $\text{Cov}\left(\eta_{t-i}, \eta_{t-j}\right) = C_t^{ij} L_{t-i} L_{t-j}^{\top}$ between times $t - i$ and $t - j$, which is no longer $0$. While this parameterization results in a non-stationary multivariate process through varying coregionalization [22, 23], a key difference is that both the coregionalization coefficient matrix $L_t$ and the temporal correlation $C_t$ are generated by a deep neural network. In this sense, our model can better characterize the empirical cross-covariance matrices of the residuals (see empirical examples in Fig. 1). As $\mu_t^{\text{bat}}$, $L_t^{\text{bat}}$, and $d_t^{\text{bat}}$ are default outputs of the base probabilistic model, we can compute the overall likelihood (with overlapped data) as

$$\mathcal{L} = \sum_{t=D}^{T} \log p\left(z_t^{\text{bat}} \mid \mu_t^{\text{bat}}, \Sigma_t^{\text{bat}}\right). \tag{10}$$

Here, computing the log-likelihood involves evaluating the inverse and the determinant of $\Sigma_t^{\text{bat}}$ with size $DB \times DB$, for which a naive implementation has a prohibitive time complexity of $\mathcal{O}\left(D^3 B^3\right)$. However, our parameterization of $\Sigma_t^{\text{bat}}$ as $E + ACA^{\top}$, where $E = \text{diag}(d_t^{\text{bat}})$, $A = L_t^{\text{bat}}$, and $C = C_t \otimes I_R$, allows us to leverage the Sherman–Morrison–Woodbury identity (matrix inversion lemma) and the companion matrix determinant lemma to simplify the computation:

$$(E + ACA^{\top})^{-1} = E^{-1} - E^{-1}A(C^{-1} + A^{\top}E^{-1}A)^{-1}A^{\top}E^{-1},$$
$$\det\left(E + ACA^{\top}\right) = \det\left(C^{-1} + A^{\top}E^{-1}A\right)\det\left(C\right)\det\left(E\right). \tag{11}$$

Then, the likelihood calculation only requires computing the inverse and determinant of a $DR \times DR$ matrix, specifically $C^{-1} + A^{\top}E^{-1}A$. These computations can be efficiently performed using Cholesky factorization. Detailed computations are provided in Appendix §A.2.

Modeling the latent process $r_t$ offers several advantages. Firstly, because $r_t$ has a much lower dimension than $\varepsilon_t$, modeling the cross-correlation of $r_t$ results in a significantly smaller $DR \times DR$ covariance matrix compared to the $DB \times DB$ covariance matrix of $\varepsilon_t$. Secondly, since $r_t$ follows an isotropic Gaussian distribution, the covariance of $r_t^{\mathrm{bat}}$ can be parameterized with a Kronecker structure $C_t \otimes I_R$. This greatly simplifies the task into learning a $D \times D$ correlation matrix shared by all time series in a batch. Lastly, similar to GPVar, we can still train the model in an end-to-end manner using a subset of time series in each iteration to ensure computational efficiency (Fig. 3).

## 4.2 Multistep-ahead Rolling Prediction

Autoregressive models perform multistep-ahead forecasting in an iterative manner, where the model generates a sample at each time step during prediction, using it as input for the subsequent step, and continuing this process until the desired prediction range is reached. Our approach enhances this process, similar to Zheng et al. [9], by offering additional calibration based on the learned correlation matrix $C_t$. Assuming observations are available up to time step $t$, the conditional distribution of $\eta_{t+1}$ given errors in the past $(D-1)$ steps, can be derived as

$$\eta_{t+1} \mid \eta_t, \eta_{t-1}, \ldots, \eta_{t-D+2} \sim \mathcal{N}\left(\Sigma_* \Sigma_{\mathrm{obs}}^{-1} \eta_{\mathrm{obs}}, \Sigma_{t+1} - \Sigma_* \Sigma_{\mathrm{obs}}^{-1} \Sigma_*^\top\right), \qquad (12)$$

where $\eta_{\mathrm{obs}} = \mathrm{vec}\left(\left[\eta_{t-D+2}, \ldots, \eta_{t-1}, \eta_t\right]\right) \in \mathbb{R}^{(D-1)B}$ represents the set of residuals, accessible at forecasting step $t+1$. Here, $\Sigma_{\mathrm{obs}}$ is a $(D-1)B \times (D-1)B$ partition of $\Sigma_{t+1}^{\mathrm{bat}}$ that captures the covariance of $\eta_{\mathrm{obs}}$, and $\Sigma_*$ is a $B \times (D-1)B$ partition of $\Sigma_{t+1}^{\mathrm{bat}}$ representing the covariance between $\eta_{t+1}$ and $\eta_{\mathrm{obs}}$, i.e., $\Sigma_{t+1}^{\mathrm{bat}} = \begin{bmatrix} \Sigma_{\mathrm{obs}} & \Sigma_*^\top \\ \Sigma_* & \Sigma_{t+1} \end{bmatrix}$. For conciseness, we omit the time index $t$ in $\Sigma_{\mathrm{obs}}, \Sigma_*$ and $\eta_{\mathrm{obs}}$. Since $\mu_{t+1}$ is a deterministic output from the base model, a sample of the target variables $\tilde{z}_{t+1}$ can be derived by first drawing a sample $\tilde{\eta}_{t+1}$ from Eq. (12), then combining it with the predicted mean vector $\mu_{t+1}$ as $\tilde{z}_{t+1} = \mu_{t+1} + \tilde{\eta}_{t+1}$. It should be noted that we can still leverage the Sherman-Morrison-Woodbury identity when computing the inverse $\Sigma_{\mathrm{obs}}^{-1}$.

By taking the sample $\tilde{\eta}_{t+1}$ as an observed residual, we can iteratively apply the process described in Eq. (12) to derive a trajectory of $\{\tilde{z}_{t+q}\}_{q=1}^Q$. Repeating this procedure allows us to generate multiple samples, characterizing the predictive distribution at each time step.

# 5 Experiments

## 5.1 Evaluation of Predictive Performance

**Datasets**. We use widely recognized time series benchmarking datasets from GluonTS [24]. The prediction range $(Q)$ for each dataset follows the configurations provided by GluonTS. We applied a sequential split into training, validation, and testing sets for each dataset. Each dataset was standardized using the mean and standard deviation from the training set, and predictions were rescaled to their original values for evaluation. Further details on the datasets can be found in Appendix §A.1.

**Base probabilistic models**. We integrated the proposed method into two distinct autoregressive models: the RNN-based GPVar [3] and the decoder-only Transformer [25]. These models are trained to generate distribution parameters as described in §4. Our approach can be applied to other autoregressive multivariate models with minimal adjustments, provided the final prediction follows a multivariate Gaussian distribution. The implementation is based on using PyTorch Forecasting [26]. Both models use lagged time series values and additional features or covariates as inputs. Details on model training (§A.3), hyperparameter tuning (§A.5), and the base model (§A.6) are provided in Appendix §A. The code is available at https://github.com/rottenivy/mv_pts_correlatederr.

**Dynamic correlation matrix**. We introduce a limited number of additional parameters to project the state vector $\mathbf{h}_t$ into component weights $w_{m,t}$, which are used to generate the dynamic correlation matrix $C_t$. The number of base kernels $(M)$ for generating $C_t$ and the associated lengthscale set $\{l_m\}_{m=1}^{M-1}$ are treated as hyperparameters. We perform a grid search over $M = 2, 3, 4$ and two sets of lengthscales—$\{0.5, 1.5, \ldots\}$ and $\{1.0, 2.0, \ldots\}$. Models with the best validation loss are selected. These different lengthscales capture varying correlation decay rates, enabling the model to account

Table 1: $\text{CRPS}_{\text{sum}}$ accuracy comparison. "w/o" denotes methods without time-dependent errors, while "w/" indicates our method. Bold values show models with time-dependent errors performing better. Mean and standard deviation are obtained from 10 runs of each model.

| | VAR | GARCH | GPVar | | Transformer | |
|---|---|---|---|---|---|---|
| | | | w/o | w/ | w/o | w/ |
| exchange_rate | 0.0033±0.0000 | 0.0435±0.0001 | 0.0068±0.0004 | 0.0117±0.0004 | 0.0055±0.0002 | **0.0042±0.0002** |
| solar | 0.7663±0.0050 | 0.8752±0.0015 | 0.7103±0.0065 | **0.6929±0.0039** | 0.4960±0.0034 | **0.4132±0.0027** |
| electricity | 0.1264±0.0006 | 0.2847±0.0015 | 0.0430±0.0005 | **0.0403±0.0004** | 0.0494±0.0004 | 0.0638±0.0003 |
| traffic | 3.5241±0.0084 | 0.4459±0.0005 | 0.1095±0.0002 | **0.0649±0.0002** | 0.0717±0.0002 | 0.0981±0.0002 |
| wikipedia | 26.2025±0.0389 | 0.6699±0.0045 | 0.1745±0.0008 | **0.0743±0.0009** | 0.0841±0.0013 | **0.0500±0.0005** |
| m4_hourly | 0.2352±0.0008 | 0.2758±0.0006 | 0.0613±0.0004 | **0.0358±0.0002** | 0.0651±0.0004 | **0.0616±0.0003** |
| m1_quarterly | N/A | N/A | 0.3942±0.0030 | **0.3538±0.0017** | 0.4448±0.0027 | **0.4367±0.0028** |
| pems03 | 0.0598±0.0002 | 0.3202±0.0007 | 0.0503±0.0001 | **0.0491±0.0002** | 0.0490±0.0001 | **0.0386±0.0001** |
| uber_hourly | N/A | N/A | 0.0342±0.0006 | **0.0222±0.0004** | 0.0632±0.0003 | **0.0513±0.0005** |
| | | | avg. rel. impr. | 13.79% | avg. rel. impr. | 6.91% |

for different temporal patterns. The time-varying component weights enable dynamic adaptation to changing correlation structures over time.

**Baselines**. We evaluate the proposed method by comparing it with a baseline model trained without accounting for error cross-correlation (Eq. (4)). The baseline model represents a special case of our model with $\boldsymbol{C}_t = \boldsymbol{I}_D$. To ensure a straightforward and fair comparison, we align the cross-correlation range ($D$) with the prediction range ($Q$), ensuring identical data sampling processes for both methods. Additionally, we set $P = Q$ following the default configuration in GluonTS. We also include VAR and GARCH as naive baseline models (see Appendix §A.4).

**Metrics**. We use the Continuous Ranked Probability Score (CRPS) [27] as the main metric:

$$\text{CRPS}(F, z) = \mathbb{E}_F |Z - z| - \frac{1}{2} \mathbb{E}_F |Z - Z'|, \tag{13}$$

where $F$ is the cumulative distribution function (CDF) of the predicted variable, $z$ is the observation, $Z$ and $Z'$ are independent copies of the prediction samples associated with the distribution $F$. To evaluate multivariate dependencies in the time series data, we compute $\text{CRPS}_{\text{sum}}$ by first summing both the forecast and ground-truth values across all time series and then calculating the CRPS over the resulting sums [3, 16, 17]. As $\text{CRPS}_{\text{sum}}$ may overlook model performance on individual dimensions [28], we also report additional metrics, e.g., the energy score [27, 29], in Appendix §B.1.

**Training dynamics**. Our approach incurs additional training costs per optimization step due to the more complex likelihood function. As shown in Appendix §B.3, the training time per epoch for models using our method is generally longer than that of baseline methods. However, our parameterization allows for scalability to large time series datasets by using a small random subset of time series at each optimization step during training.

**Benchmark results**. The $\text{CRPS}_{\text{sum}}$ results are presented in Table 1. Our method achieves an average improvement of 13.79% for GPVar and 6.91% for the Transformer model. It is important to note that the degree of performance enhancement varies across different base models and datasets, influenced by factors such as the inherent data characteristics and the performance of different model architectures. The alignment between the actual correlation structure and our kernel assumption also plays a crucial role in the effectiveness of our method. Additionally, our approach demonstrates consistent improvements across five different metrics, with significant gains in multivariate metrics such as the energy score (Appendix §B.2).

To provide further insights, we compare the residual autocorrelation and cross-lag correlation with and without applying our method in Appendix §B.5.1, showing that our method effectively reduces cross-correlations in many scenarios. We use ACF plot comparisons to illustrate the reduction in autocorrelation and cross-correlation plot comparisons to demonstrate the decrease in cross-lag correlation. The residuals generated by the model with our method exhibit weaker cross-correlations, which is particularly enhanced by the calibration process during prediction (§4.2).

Furthermore, Appendix §B.5.2 separates the accuracy improvement over forecast steps for each dataset. The performance improvement is shown to be related to both the absolute time across the

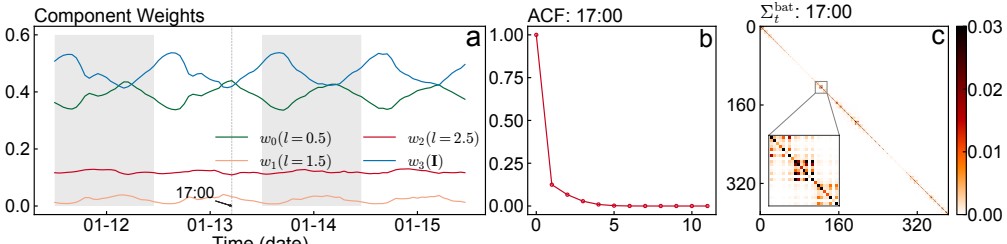

Figure 4: (a) Component weights for generating $C_t$ for a batch of time series ($B = 8$) from the `m4_hourly` dataset obtained by the GPVar model. Parameters $w_0, w_1, w_2$ represent the component weights of the kernel matrices associated with lengthscales $l = 0.5, 1.5, 2.5$, and $w_3$ is the component weight of the identity matrix. Shaded areas distinguish different days; (b) The autocorrelation function (ACF) indicated by the correlation matrix $C_t$ at 17:00. Given the rapid decay of the ACF, we only plot 12 lags to enhance visualization; (c) The corresponding covariance matrix of the associated target variables $\Sigma_t^{\text{bat}}$ at 17:00. A zoom-in view of a $3B \times 3B$ region is illustrated in the plot, where the diagonal blocks represent $B \times B$ covariance matrices $\Sigma_{t'}$ of $\mathbf{z}_{t'}$ over three consecutive time steps. The off-diagonal blocks describe the cross-covariance $\text{Cov}(\mathbf{z}_{t-\Delta}, \mathbf{z}_t), \forall \Delta \neq 0$. For visualization clarity, covariance values are clipped to the range $[0, 0.03]$.

temporal span of the dataset (especially for time series with strong periodic patterns) and the relative time over the prediction horizon.

## 5.2 Model Interpretation

Our method captures error cross-correlation through the dynamic construction of a covariance matrix, achieved by combining kernel matrices with varying lengthscales in a dynamically weighted sum. A small lengthscale corresponds to short-range positive correlations, while a large lengthscale captures positive correlations over longer lags.

In Fig. 4, we depict the dynamic component weights and the resulting autocorrelation function (the first row of the correlation matrix $C_t$) for a batch of time series from the `m4_hourly` dataset spanning a four-day window. We also provide the covariance matrix of $\mathbf{z}_t^{\text{bat}}$ using the correlation matrix and model outputs at a specific time of day. The component weight $w_3$, corresponding to the identity matrix, dominates throughout the observation period. This suggests that the error correlation is generally mild over time. This behavior is influenced by the Kronecker structure used to parameterize the covariance over the low-dimensional latent variables $\mathbf{r}_t$, which assumes all latent processes share the same autocorrelation structure. Given the Kronecker structure, the model tends to learn the mildest temporal correlation among the time series in a batch.

Moreover, we observe that the dynamic component weights adjust the correlation strengths. Specifically, when the weight assigned to the identity matrix ($w_3$) increases, the error process tends to be more independent. In contrast, when the weights assigned to the other kernel matrices ($w_0$, $w_1$, and $w_2$) are larger, the error process becomes more correlated, as the kernel matrices with different lengthscales combine to formulate a specific correlation structure. Fig. 4(a) demonstrates pronounced daily patterns in temporal correlation, particularly when errors exhibit increased correlation around 17:00 each day. The corresponding autocorrelation function is shown in Fig. 4(b). Fig. 4(c) illustrates the corresponding covariance matrix of the associated target variables within the cross-correlation horizon. The diagonal blocks represent the contemporaneous covariance $\Sigma_t$ of $\mathbf{z}_t$ at each time step, while the off-diagonal blocks capture the cross-covariance $\text{Cov}(\mathbf{z}_{t-\Delta}, \mathbf{z}_t)$ for $\forall \Delta \neq 0$, effectively modeled by our approach. The zoomed-in view provides a $3B \times 3B$ region that illustrates the cross-covariance within two lags. We observe that the cross-covariance is most pronounced at lag 1, consistent with the observation in Fig. 4(a) that the component weight $w_0$, assigned to the base kernel matrix with lengthscale $l = 0.5$, is more pronounced than $w_1$ and $w_2$.

## 6 Discussion

In this section, we discuss factors that influence the performance of our method. Specifically, we highlight the effectiveness of our model in long-term forecasting across various scenarios. We also

discuss the effect of scaling up to larger batch sizes during prediction. Additionally, we examine the impact of non-Gaussian errors on model performance.

**Long-term forecasting**. The advantage of modeling error correlation can vary in long-term forecasting, especially in autoregressive predictions where errors accumulate and propagate over time. Using residuals from previous time steps to calibrate forecasts may be beneficial for non-stationary segments of the time series. However, for time series with strong periodic effects, the model may also rely on seasonal lags. As shown in Fig. 21 and Fig. 22 of the Appendix, the advantage of modeling error correlation can decrease in longer-term forecasts compared to shorter-term forecasts for some datasets with strong periodic effects (e.g., the `traffic` dataset in Fig. 21). It is not necessarily true that the advantage diminishes for long-horizon predictions, as the effectiveness of our method depends on the quality of predictions during inference. In cases where the model provides accurate long-term forecasts, the benefit of modeling correlated errors may be less pronounced.

**Scalability**. Increasing the number of time series $B$ in a batch leads to higher training costs. Because the model requires numerous iterations over the dataset for optimization, using a large $B$ during training is not feasible. However, during prediction, the batch size can be increased to leverage more information. This may enhance both prediction accuracy and error calibration, provided sufficient memory is available. We demonstrate the effect of increasing batch size during prediction in Appendix §B.4 through additional experiments. Both models, with and without our method, show improvement from increased batch sizes during prediction, as reflected by a decrease in $\mathrm{CRPS}_{\mathrm{sum}}$.

**Non-Gaussian errors**. For the baseline model, assuming Gaussian errors may lead to model misspecification, resulting in more correlated residuals. To address this issue, we also trained the baseline models using the likelihood of a multivariate $t$-distribution; the results are presented in Table 15 of the Appendix. Although using an alternative distribution can lead to better performance on some datasets without our method, we observed that our method effectively closes the performance gap when the $t$-distribution outperforms the Gaussian assumption. We chose the Gaussian distribution for its beneficial properties, including its marginalization rule and well-defined conditional distributions, both essential for statistically consistent model training and reliable inference. Thus, a more effective approach could involve first transforming the original observations into Gaussian-distributed data using a Gaussian Copula [3], followed by applying our method.

# 7   Conclusion and Broader Impacts

This paper presents a novel approach for addressing error cross-correlation in multivariate probabilistic time series forecasting, specifically for models with autoregressive properties and Gaussian distribution outputs. We construct a dynamic covariance matrix using a small set of independent and identically distributed latent temporal processes. These latent processes effectively model temporal correlation and integrate seamlessly into the base model, where the contemporaneous covariance is parameterized by a low-rank-plus-diagonal structure. This approach enables the modeling and prediction of a time-varying covariance matrix for the target time series variables. The experimental results demonstrate its effectiveness in enhancing uncertainty quantification.

Our contributions are two-fold. First, our approach relaxes the time-independent error assumption during the training process for probabilistic forecasting models, addressing the reality that residuals are typically time-dependent. Second, the learned cross-correlation improves multistep-ahead predictions by refining the distribution output at each forecasting step. These enhancements to existing models have broader implications for fields such as finance, healthcare, and energy, where improved forecasts and uncertainty quantification can lead to more informed decisions.

There are several avenues for future research. First, the Kronecker structure $C_t \otimes \mathbf{I}_R$ for the covariance matrix of the latent variable $r_t^{\mathrm{bat}}$ may be too restrictive for multivariate time series problems. Exploring more flexible covariance structures, such as employing different $C_{r,t}$ matrices for each latent temporal process as in the linear model of coregionalization (LMC, [22, 23]), could be a promising direction for further investigation. Second, the parameterization of $C_t$ could be expanded. Instead of using SE kernels, $C_t$ could be parameterized as fully learnable positive definite symmetric Toeplitz matrices. For example, an $\mathrm{AR}(p)$ process has a covariance structure in Toeplitz form, allowing for the modeling of negative correlations. This alternative approach could offer greater flexibility in capturing complex correlation patterns in multivariate time series data.

## Acknowledgments and Disclosure of Funding

We acknowledge the support from the Natural Sciences and Engineering Research Council (NSERC) of Canada (Discovery Grant). Vincent Zhihao Zheng also acknowledges the support received from the FRQNT B2X Doctoral Scholarship Program.

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

# Appendix

**Table of Contents**

# A   Experimental Details

## A.1   Datasets

We performed experiments on a diverse set of real-world datasets obtained from GluonTS [24]. These datasets include:

- `electricity` [30]: Hourly electricity consumption data collected from a total of 370 households over the period spanning from January 2012 to June 2014.
- `m4_hourly` [31]: Hourly time series data from various domains, covering microeconomics, macroeconomics, finance, industry, demographics, and various other fields, are sourced from the M4-competition.
- `exchange_rate` [32]: Daily exchange rate information for eight different countries spanning the period from 1990 to 2016.
- `m1_quarterly` [33]: Quarterly time series data spanning seven different domains.
- `pems03` [34]: Traffic flow records obtained from Caltrans District 3 and accessed through the Caltrans Performance Measurement System (PeMS). The records are aggregated at a 5-minute interval.

- `solar` [32]: Hourly time series representing solar power production data in the state of Alabama for the year 2006.
- `traffic` [35]: Hourly traffic occupancy rates recorded by sensors installed in the San Francisco freeway system between January 2008 and June 2008.
- `uber_hourly` [36]: Hourly time series of Uber pickups in New York City spanning from February to July 2015.
- `wikipedia` [37]: Daily page views for 2,000 Wikipedia pages spanning from January 2012 to March 2014.

These datasets are widely employed for benchmarking time series forecasting models, following their default configurations in GluonTS, including granularity, prediction range ($Q$), and the number of rolling evaluations. For each dataset, we performed a sequential split into training, validation, and testing sets, with the temporal length of the validation set matching that of the testing set. The temporal length of the testing set was determined by considering the prediction range and the required number of rolling evaluations. For instance, the testing horizon for the `traffic` dataset is computed as $24 + 7 - 1 = 30$ time steps. As a result, the model will predict 24 steps ($Q$) sequentially, with 7 distinct consecutive prediction start timestamps, also known as 7 forecast instances. In our experiments, we align the conditioning range ($P$) with the prediction range ($Q$), maintaining consistency with the default setting in GluonTS. For simplicity, we set the autocorrelation horizon ($D$) to also match the prediction range ($Q$). Essentially, in this paper, we have $P = Q = D$. Each dataset was standardized using the mean and standard deviation from the training set. Predictions were rescaled to their original values for computing evaluation metrics. The statistics of all datasets are summarized in Table 2.

Table 2: Dataset summary.

| Dataset | Granularity | # of time series | # of time steps | $Q$ | Rolling evaluation |
|---|---|---|---|---|---|
| electricity | hourly | 370 | 5,857 | 24 | 7 |
| m4_hourly | hourly | 414 | 1,008 | 48 | 7 |
| exchange_rate | workday | 8 | 6,101 | 30 | 5 |
| m1_quarterly | quarterly | 281 | 48 | 8 | 1 |
| pems03 | 5min | 358 | 26,208 | 12 | 24 |
| solar | hourly | 137 | 7,033 | 24 | 7 |
| traffic | hourly | 963 | 4,025 | 24 | 7 |
| uber_hourly | hourly | 262 | 8,343 | 24 | 7 |
| wikipedia | daily | 2,000 | 792 | 30 | 5 |

### A.2 Multivariate Likelihood with Correlated Errors

The probability density function of a multivariate normal distribution with autocorrelated errors, as described in §4.1, is defined in Eq. (14). For simplicity, we omit the subscript $t$ and superscript bat for all notations:

$$f(\boldsymbol{z}) = (2\pi)^{-B/2} |\boldsymbol{\Sigma}|^{-1/2} \exp\left(-\frac{1}{2} (\boldsymbol{z} - \boldsymbol{\mu})^\top \boldsymbol{\Sigma}^{-1} (\boldsymbol{z} - \boldsymbol{\mu})\right). \tag{14}$$

We use the negative log likelihood (NLL) of an observed $\boldsymbol{z}$ as the loss function for training our model. The NLL can be calculated as the negative log of the probability density function in Eq. (14):

$$\mathcal{L}_{NLL} = -\ln L(\boldsymbol{z}) = \frac{1}{2}\left[\ln|\boldsymbol{\Sigma}| + (\boldsymbol{z} - \boldsymbol{\mu})^\top \boldsymbol{\Sigma}^{-1} (\boldsymbol{z} - \boldsymbol{\mu}) + B \ln(2\pi)\right], \tag{15}$$

where $B$ is the number of time series in a batch. The covariance matrix is parameterized as $\boldsymbol{\Sigma} = \boldsymbol{L}(\boldsymbol{C} \otimes \mathbf{I}_R)\boldsymbol{L}^\top + \boldsymbol{E}$. In this parameterization, $\boldsymbol{L} \in \mathbb{R}^{DB \times DR}$ is the covariance factor, $\boldsymbol{C} \in \mathbb{R}^{D \times D}$ is the autocorrelation matrix, $\boldsymbol{E} = \mathrm{diag}(\boldsymbol{d})$, and $\boldsymbol{d} \in \mathbb{R}_+^{DB}$ are the diagonal elements. The bottleneck in evaluating this NLL lies in the calculation of the inverse and determinant of $\boldsymbol{\Sigma}$. Therefore, we can simplify the calculation using the Sherman–Morrison–Woodbury identity (matrix inversion lemma) and the companion matrix determinant lemma:

$$\begin{aligned} \boldsymbol{\Sigma}^{-1} &= \left(\boldsymbol{E} + \boldsymbol{L}(\boldsymbol{C} \otimes \mathbf{I}_R)\boldsymbol{L}^\top\right)^{-1} \\ &= \boldsymbol{E}^{-1} - \boldsymbol{E}^{-1}\boldsymbol{L}\left((\boldsymbol{C} \otimes \mathbf{I}_R)^{-1} + \boldsymbol{L}^\top \boldsymbol{E}^{-1}\boldsymbol{L}\right)^{-1}\boldsymbol{L}^\top \boldsymbol{E}^{-1}. \end{aligned} \tag{16}$$

Consequently, the Mahalanobis term in Eq. (15) becomes:

$$
\begin{aligned}
\boldsymbol{\eta}^\top \boldsymbol{\Sigma}^{-1} \boldsymbol{\eta} =& \boldsymbol{\eta}^\top \boldsymbol{E}^{-1} \boldsymbol{\eta} - \boldsymbol{\eta}^\top \boldsymbol{E}^{-1} \boldsymbol{L} \left( (\boldsymbol{C} \otimes \mathbf{I}_R)^{-1} + \boldsymbol{L}^\top \boldsymbol{E}^{-1} \boldsymbol{L} \right)^{-1} \boldsymbol{L}^\top \boldsymbol{E}^{-1} \boldsymbol{\eta} \\
=& \boldsymbol{\eta}^\top \boldsymbol{E}^{-1} \boldsymbol{\eta} - \boldsymbol{\eta}^\top \boldsymbol{E}^{-1} \boldsymbol{L} \left( \boldsymbol{L}_{cap} \boldsymbol{L}_{cap}^\top \right)^{-1} \boldsymbol{L}^\top \boldsymbol{E}^{-1} \boldsymbol{\eta} \\
=& \boldsymbol{\eta}^\top \boldsymbol{E}^{-1} \boldsymbol{\eta} - \left( \boldsymbol{L}_{cap}^{-1} \boldsymbol{L}^\top \boldsymbol{E}^{-1} \boldsymbol{\eta} \right)^\top \left( \boldsymbol{L}_{cap}^{-1} \boldsymbol{L}^\top \boldsymbol{E}^{-1} \boldsymbol{\eta} \right) \\
=& \boldsymbol{\eta}^\top \boldsymbol{E}^{-1} \boldsymbol{\eta} - \boldsymbol{k}^\top \boldsymbol{k},
\end{aligned}
\tag{17}
$$

where $\boldsymbol{k} = \boldsymbol{L}_{cap}^{-1} \boldsymbol{L}^\top \boldsymbol{E}^{-1} \boldsymbol{\eta}$. $\boldsymbol{L}_{cap}$ is the Cholesky factor of the capacitance matrix $\left( (\boldsymbol{C} \otimes \mathbf{I}_R)^{-1} + \boldsymbol{L}^\top \boldsymbol{E}^{-1} \boldsymbol{L} \right)$. The computation of $\boldsymbol{k}$ can be efficiently resolved by solving the linear system of equations $\boldsymbol{L}_{cap} \boldsymbol{k} = \boldsymbol{L}^\top \boldsymbol{E}^{-1} \boldsymbol{\eta}$. Since $\boldsymbol{E}$ is a diagonal matrix, the only matrix inverse we need to calculate in Eq. (17) is $(\boldsymbol{C} \otimes \mathbf{I}_R)^{-1}$, which can be further simplified as $\boldsymbol{C}^{-1} \otimes \mathbf{I}_R$. Recall that $\boldsymbol{C}$ is a $D \times D$ autocorrelation matrix. Therefore, calculating its inverse is much easier than computing the inverse of $\boldsymbol{\Sigma}$, which is a $DB \times DB$ matrix. Moreover, the computational cost does not scale with the number of time series $B$ in a batch.

The calculation of the determinant can also be greatly simplified with our parameterization:

$$
\begin{aligned}
\ln|\boldsymbol{\Sigma}| &= \ln|\boldsymbol{E} + \boldsymbol{L} \left( \boldsymbol{C} \otimes \mathbf{I}_R \right) \boldsymbol{L}^\top| \\
&= \ln|(\boldsymbol{C} \otimes \mathbf{I}_R)^{-1} + \boldsymbol{L}^\top \boldsymbol{E}^{-1} \boldsymbol{L}| + \ln|\boldsymbol{C} \otimes \mathbf{I}_R| + \ln|\boldsymbol{E}| \\
&= 2 \sum_i^{DR} \ln \left[ \boldsymbol{L}_{cap} \right]_{i,i} + 2R \sum_i^{D} \ln \left[ \boldsymbol{L}_C \right]_{i,i} + \sum_i^{DB} \ln \left[ \boldsymbol{E} \right]_{i,i},
\end{aligned}
\tag{18}
$$

where $\boldsymbol{L}_C$ is the Cholesky factor of the autocorrelation matrix $\boldsymbol{C}$.

### A.3 Training Procedure

**Compute used** All models in the paper were trained in an Anaconda environment with access to one AMD Ryzen Threadripper PRO 5955WX CPU and four NVIDIA RTX A5000 GPUs (each with 24 GB of memory).

**Batch size** We adopt the approach of GPVar [3] by using $B = 20$ time series in a sample slice and a batch size of 16. Because our data sampler selects one slice of time series as a batch instead of sampling 16 slices simultaneously, we set `accumulate_grad_batches` to 16 to achieve an effective batch size of 16.

**Training loop** Each epoch involves training the model on up to 400 batches from the training set, followed by computing the NLL on the validation set. Training stops when any of the following conditions are met:

- A total of 10,000 gradient updates have been performed during model training,
- No improvement in the best NLL value on the validation set is observed for 10 consecutive epochs.

We select the version of the model that achieved the best NLL value on the validation set.

### A.4 Naive Baseline Description

In this paper, we employ VAR [38] (Vector Autoregression) and GARCH [39] (Generalized Autoregressive Conditionally Heteroskedasticity) as two naive baseline models. The VAR($p$) model is defined as

$$
\mathbf{z}_t = \mathbf{c} + \boldsymbol{A}_1 \mathbf{z}_{t-1} + \cdots + \boldsymbol{A}_p \mathbf{z}_{t-p} + \boldsymbol{\epsilon}_t, \boldsymbol{\epsilon}_t \sim \mathcal{N}(\mathbf{0}, \boldsymbol{\Sigma}_\epsilon),
\tag{19}
$$

where $A_i$ is an $N \times N$ coefficient matrix, and $\mathbf{c}$ is the intercept. We use a VAR model of lag 1 (i.e., a VAR(1) model) in the experiments. The parameters of Eq. (19) are estimated using ordinary least squares (OLS), following the procedure in [38].

The GARCH model describes the conditional covariance matrix of the error term in a multivariate system. Suppose the model for the conditional mean is an AR(1) model:

$$
\mathbf{z}_t = \mathbf{c} + \boldsymbol{A}_1 \mathbf{z}_{t-1} + \boldsymbol{\epsilon}_t,
\tag{20}
$$

where the error term is modeled as

$$\boldsymbol{\epsilon}_t = \boldsymbol{H}_t^{1/2}\mathbf{e}_t, \tag{21}$$

where $\boldsymbol{H}_t$ is an $N \times N$ conditional covariance matrix, and $\mathbf{e}_t$ is an $N \times 1$ standard normal vector, $\mathbf{e}_t \sim \mathcal{N}(\mathbf{0}, \mathbf{I}_N)$. In the experiments, we use the DCC-GARCH$(1, 1)$ model [40], where the conditional covariance matrix $\boldsymbol{H}_t$ is defined as

$$\boldsymbol{H}_t = \boldsymbol{D}_t \boldsymbol{R}_t \boldsymbol{D}_t, \tag{22}$$

where $\boldsymbol{D}_t = \mathrm{diag}\,(\mathbf{h}_t)^{1/2}$, and $\mathbf{h}_t$ contains the variances for each time series. $\boldsymbol{R}_t$ is the conditional correlation matrix in the DCC-GARCH model. The parameters of the DCC-GARCH model are estimated with the log-likelihood function:

$$\mathcal{L} = -\frac{1}{2}\sum_{t=1}^{T}\left[N \ln(2\pi) + 2\ln|\boldsymbol{D}_t| + \ln|\boldsymbol{R}_t| + \mathbf{e}_t^\top \boldsymbol{R}_t^{-1}\mathbf{e}_t\right]. \tag{23}$$

In this paper, we implement the VAR model using `statsmodels` [41] and the DCC-GARCH model using `mgarch` [42].

## A.5 Hyperparameter Search

The hyperparameters and training configuration largely align with those used in the GPVar paper [3]. All DL models are trained using the Adam optimizer with $l2$ regularization set to 1e-8, and gradients are clipped at 10.0. For all methods, we limit the total number of gradient updates to 10,000 and decay the learning rate by a factor of 2 after 500 consecutive updates without improvement. Table 3 lists the parameters that are tuned, as well as the hyperparameters that are kept constant across all datasets and not subject to tuning.

Table 3: Hyperparameters values that are fixed or searched over a range during hyperparameter tuning.

| Hyperparameter | Value or Range Searched |
| --- | --- |
| learning rate | [1e-4, 1e-3, 1e-2] |
| LSTM cells / `d_model` of Transformer | [10, 20, 40] |
| LSTM layers / Transformer decoder layers | 2 |
| `n_heads` (Transformer) | 2 |
| rank | 10 |
| sampling dimension | 20 |
| dropout | 0.01 |
| batch size | 16 |

To tune the hyperparameters of each model, we conduct a grid search over nine parameters on each dataset. The best hyperparameters for each base model–dataset combination are selected based on the lowest validation loss. Once the optimal learning rate and hidden size are determined, we apply the same hyperparameters to models both with and without our method.

The number of base kernels ($M$) for generating $\boldsymbol{C}_t$ and the associated lengthscale set $\{l_m\}_{m=1}^{M-1}$ are two additional hyperparameters when applying our method. The optimal values of $M$ and $\{l_m\}_{m=1}^{M-1}$ are selected in a similar manner via hyperparameter search. The values of $M$ and $\{l_m\}_{m=1}^{M-1}$ explored during hyperparameter tuning are shown in Table 4. There are six possible combinations. For example, if we set $M = 3$ and choose the initial lengthscale to be 1.0, the lengthscales for generating the component kernels will be $\{1.0, 2.0\}$ since the last weight corresponds to the identity matrix.

## A.6 Base Model Description and Input Features

The input to the base models consists of lagged time series values and generic features that encode time and identify each time series. The number of lagged values used is determined by the time-frequency of each dataset. Specifically, we use lags [1, 24, 168] for hourly data; [1, 7, 14] for daily data; and [1, 2, 4, 12, 24, 48] for data with a granularity of less than one hour. For all other datasets, we only use the lag-1 values.

Table 4: Hyperparameters values of our method that are searched over a range during hyperparameter tuning.

| Hyperparameter | Value or Range Searched |
|---|---|
| number of kernels $M$ | [2, 3, 4] |
| possible lengthscales $\{l_m\}_{m=1}^{M-1}$ | $[\{0.5, 1.5, \dots\}, \{1.0, 2.0, \dots\}]$ |

We use generic features to represent time. For datasets with a granularity of one hour or less, we include features for the hour of the day and the day of the week. For daily datasets, we use the day of the week feature. Additionally, each time series is distinguished by an identifier number. All features are encoded with a single value; for example, the hour of the day feature takes values in [0, 23]. These feature values are concatenated with the RNN or Transformer input at each time step to generate the model input vector $\mathbf{y}_t$.

As illustrated in §4, our method requires a state vector $\mathbf{h}_t$ at each time step to generate the parameters for the predictive distribution and the dynamic weights for correlation matrix kernels. We use two different neural architectures for this purpose: RNN and Transformer, both of which preserve autoregressive properties. Specifically, we use an LSTM as our base model for the RNN and a decoder-only Transformer (i.e., the GPT model [25]) for the Transformer. Table 5 and Table 6 summarize the number of parameters for the GPVar and Transformer models across each dataset.

Table 5: Number of parameters of the GPVar model for each dataset.

|  | covariate embedding | rnn | distribution proj | covariance proj (our method) |
|---|---|---|---|---|
| exchange_rate | 60 | 6.1k | 252 | 84 |
| solar | 3.7k | 26.6k | 492 | 164 |
| electricity | 16.5k | 29.6k | 492 | 164 |
| traffic | 72.5k | 34.6k | 492 | 164 |
| wikipedia | 200k | 5.7k | 132 | 44 |
| m4_hourly | 19.7k | 10.2k | 252 | 84 |
| m1_quarterly | 6.3k | 25k | 492 | 164 |
| pems03 | 26.4k | 34.6k | 492 | 164 |
| uber_hourly | 9.7k | 28.3k | 492 | 164 |

Table 6: Number of parameters of the Transformer model for each dataset.

|  | target proj | covariate proj | covariate embedding | transformer | distribution proj | covariance proj (our method) |
|---|---|---|---|---|---|---|
| exchange_rate | 160 | 400 | 60 | 26.5k | 492 | 164 |
| solar | 40 | 400 | 3.7k | 1.8k | 132 | 44 |
| electricity | 160 | 2.4k | 16.5k | 26.5k | 492 | 164 |
| traffic | 80 | 1.8k | 72.5k | 6.8k | 252 | 84 |
| wikipedia | 160 | 4.2k | 200k | 26.5k | 492 | 164 |
| m4_hourly | 80 | 1.2k | 19.7k | 6.8k | 252 | 84 |
| m1_quarterly | 80 | 1.3k | 6.3k | 26.5k | 492 | 164 |
| pems03 | 70 | 870 | 26.4k | 1.8k | 132 | 44 |
| uber_hourly | 160 | 2k | 9.7k | 26.5k | 492 | 164 |

LSTM, a type of RNN architecture, is designed to model sequences and time series data. Unlike traditional RNNs, LSTMs can learn long-term dependencies, making them effective for tasks requiring context and memory over long sequences. A decoder-only Transformer is primarily used for sequence generation tasks, such as text generation, language modeling, and machine translation. It is a simplified version of the original Transformer model introduced by Vaswani et al. [43], consisting of only the decoder component. The LSTM model can be formulated as

$$\begin{aligned}
\mathbf{f}_t &= \sigma(\mathbf{W}_f \cdot [\mathbf{h}_{t-1}, \mathbf{y}_t] + \mathbf{b}_f), \\
\mathbf{i}_t &= \sigma(\mathbf{W}_i \cdot [\mathbf{h}_{t-1}, \mathbf{y}_t] + \mathbf{b}_i), \\
\tilde{\mathbf{C}}_t &= \tanh(\mathbf{W}_C \cdot [\mathbf{h}_{t-1}, \mathbf{y}_t] + \mathbf{b}_C), \\
\mathbf{C}_t &= \mathbf{f}_t \odot \mathbf{C}_{t-1} + \mathbf{i}_t \odot \tilde{\mathbf{C}}_t, \\
\mathbf{o}_t &= \sigma(\mathbf{W}_o \cdot [\mathbf{h}_{t-1}, \mathbf{y}_t] + \mathbf{b}_o), \\
\mathbf{h}_t &= \mathbf{o}_t \odot \tanh(\mathbf{C}_t),
\end{aligned} \tag{24}$$

where $\mathbf{y}_t$ is the input at each time step. The decoder-only Transformer can be formulated as

$$\begin{aligned}
\mathbf{Q} &= \mathbf{Y}_t \mathbf{W}_Q, \\
\mathbf{K} &= \mathbf{Y}_t \mathbf{W}_K, \\
\mathbf{V} &= \mathbf{Y}_t \mathbf{W}_V, \\
\mathbf{M} &= \mathrm{Mask}(\mathbf{K}), \\
\mathbf{Z} &= \mathrm{Softmax}\left(\frac{\mathbf{Q}\mathbf{K}^T}{\sqrt{d_k}} + \mathbf{M}\right)\mathbf{V}, \\
\mathbf{H}_t &= \mathrm{LayerNorm}(\mathbf{Y}_t + \mathbf{Z}), \\
\mathbf{FFN} &= \mathrm{ReLU}(\mathbf{H}_t \mathbf{W}_1 + \mathbf{b}_1)\mathbf{W}_2 + \mathbf{b}_2, \\
\mathbf{H}_t &= \mathrm{LayerNorm}(\mathbf{H}_t + \mathbf{FFN}),
\end{aligned} \tag{25}$$

where $\mathbf{H}_t$ is the output containing state vectors for all time steps, and $\mathbf{M}$ is a square causal mask for the sequence to preserve autoregressive properties.

## B  Metrics and Additional Results

### B.1  Metric Definition

In this paper, we repeated the evaluation process on the testing set ten times to compute the mean and standard deviation of all metrics. Metrics calculated in each independent evaluation are based on the average results from all forecast instances in the testing set. For example, the $\mathrm{CRPS_{sum}}$ reported for `traffic` is the average $\mathrm{CRPS_{sum}}$ of seven forecast instances in its testing set. 100 prediction samples were drawn for all evaluation processes.

#### B.1.1  Continuous Ranked Probability Score

The Continuous Ranked Probability Score (CRPS) is defined as:

$$\mathrm{CRPS}\,(F, z) = \mathbb{E}_F |Z - z| - \frac{1}{2}\mathbb{E}_F |Z - Z'|, \tag{26}$$

where $F$ is the cumulative distribution function (CDF) of the predicted variable, $z$ is the observation, $Z$ and $Z'$ are independent copies of a set of prediction samples associated with the distribution $F$. For a single forecast instance, we calculate the average CRPS across time series and over the prediction horizon:

$$\mathbb{E}_{i,t}\left[\mathrm{CRPS}\,(F_{i,t}, z_{i,t})\right], \tag{27}$$

where we use the empirical CDF to represent $F_{i,t}$ when predicting $z_{i,t}$. Since CRPS only compares a single ground-truth value to its predicted distribution, we also calculate the $\mathrm{CRPS_{sum}}$ [3, 16, 17] to assess multivariate dependencies in the time series data. $\mathrm{CRPS_{sum}}$ is computed by summing both the forecasted and ground-truth values across all time series and then calculating the CRPS over the resulting sums:

$$\mathbb{E}_t\left[\mathrm{CRPS}\left(F_t, \sum_i z_{i,t}\right)\right], \tag{28}$$

where the empirical $F_t$ is obtained by summing samples across time series.

### B.1.2 Quantile Loss

The Quantile Loss ($\rho$-risk) is another metric used in [10] to evaluate the performance of probabilistic forecasting:

$$L_\rho\left(z, \hat{z}^\rho\right) = 2\left(\hat{z}^\rho - z\right)\left((1 - \rho)\,\mathrm{I}_{\hat{z}^\rho > z} - \rho\mathrm{I}_{\hat{z}^\rho \leq z}\right), \tag{29}$$

where I is a binary indicator function that equals 1 when the condition is met, $\hat{z}^\rho$ represents the predicted $\rho$-quantile, and $z$ represents the ground truth value. The quantile loss serves as a metric to assess the accuracy of a given quantile $\rho$ from the predictive distribution. We summarize the quantile losses over the testing set across all time series segments by computing a normalized summation of these losses: $\left(\sum_{i,t} L_\rho\left(z_{i,t}, \hat{z}_{i,t}^\rho\right)\right) / \left(\sum_{i,t} z_{i,t}\right)$. In this paper, we evaluate the 0.5-risk and the 0.9-risk following Salinas et al. [10].

### B.1.3 Energy Score

The Energy Score (ES) generalizes the CRPS to evaluate distributional forecasts of a vector-valued random variable and is thus another multivariate metric used in this paper:

$$\mathrm{ES}(P, \mathbf{z}) = \mathop{\mathbb{E}}_{\mathbf{Z} \sim P}\|\mathbf{Z} - \mathbf{z}\|_2^\beta - \frac{1}{2}\mathop{\mathbb{E}}_{\substack{\mathbf{Z} \sim P \\ \mathbf{Z}' \sim P}}\|\mathbf{Z} - \mathbf{Z}'\|_2^\beta, \tag{30}$$

where $\|\mathbf{z}\|_2$ is the Euclidean norm. In this paper, we use $\beta = 1$, following [17]. Since we also want to aggregate over the prediction horizon, we calculate the Frobenius norm of the matrix $\|\mathbf{z}_{t+1:t+Q}\|_F$ in practice.

### B.1.4 Root Relative Mean Squared Error

The Root Relative Mean Squared Error (RRMSE) is a metric commonly used for point forecasts [44, 32, 45]. RRMSE is defined as:

$$\mathrm{RRMSE} = \frac{\sqrt{\sum_{t=1}^Q \|\mathbf{z}_t - \hat{\mathbf{z}}_t\|_2^2}}{\sqrt{\sum_{t=1}^Q \|\mathbf{z}_t - \bar{\mathbf{z}}\|_2^2}}, \tag{31}$$

where $\hat{\mathbf{z}}_t$ is obtained by taking the mean of our prediction samples, and $\bar{\mathbf{z}}$ is the mean value of the entire forecast instance. We use this metric to evaluate the mean prediction performance of our model.

### B.2 Results on Other Forecasting Metrics

We present the results for CRPS (Table 7), the 0.5-risk (Table 8), the 0.9-risk (Table 9), ES (Table 10), and RRMSE (Table 11). An "N/A" entry in the tables indicates that the naive baseline models could not be properly fitted to this dataset. We observe consistent performance improvements in the base models using our method across different evaluation metrics. Notably, in the multivariate metric ES, our method shows significant improvement, reducing the score by an average of 5.58% for GPVar and 3.21% for the Transformer.

### B.3 Training Dynamics

In Fig. 5 and Fig. 6, we compare the training dynamics of the base models trained with and without our method. Note that the likelihood losses of the two methods are not directly comparable, even for the same dataset and base model, due to differences in the likelihood structures. We observe that while our method introduces more complexity into the likelihood function, there is no evidence that it significantly prolongs model convergence. On the contrary, our method can speed up convergence for some datasets in terms of the training steps used. We also report the training time in Table 12.

### B.4 Effect of the Number of Time Series during Prediction

The number of time series does not impact training, as the model is trained using a random subset of $B$ time series at a time, independent of the total number of time series $N$. However, during prediction, the batch size can be increased beyond the training batch size of $B = 20$ for multistep-ahead rolling

Table 7: Comparison of CRPS accuracy. "w/o" denotes methods without time-dependent errors, while "w/" indicates our method. Boldface values indicate that models considering time-dependent errors have better performance. Mean and standard deviation are obtained from 10 runs of each model.

| | VAR | GARCH | GPVar | | Transformer | |
|---|---|---|---|---|---|---|
| | | | w/o | w/ | w/o | w/ |
| exchange_rate | 0.0070±0.0000 | 0.0438±0.0001 | 0.0171±0.0004 | 0.0141±0.0003 | 0.0092±0.0002 | 0.0081±0.0001 |
| solar | 0.9566±0.0022 | 0.9193±0.0010 | 0.7097±0.0047 | 0.7521±0.0027 | 0.5981±0.0021 | 0.5627±0.0018 |
| electricity | 0.1548±0.0003 | 0.2778±0.0010 | 0.0586±0.0004 | 0.0568±0.0002 | 0.0665±0.0003 | 0.0775±0.0001 |
| traffic | 19.9208±0.0495 | 0.4063±0.0002 | 0.1474±0.0001 | 0.1296±0.0001 | 0.1260±0.0001 | 0.1318±0.0001 |
| wiki | 334.6021±0.4936 | 3.0351±0.0048 | 0.3712±0.0003 | 0.3705±0.0004 | 0.3737±0.0003 | 0.2937±0.0002 |
| m4_hourly | 0.2837±0.0004 | 0.3567±0.0004 | 0.1174±0.0002 | 0.1237±0.0002 | 0.1306±0.0002 | 0.1189±0.0002 |
| m1_quarterly | N/A | N/A | 0.3942±0.0030 | 0.3538±0.0017 | 0.4448±0.0027 | 0.4367±0.002 |
| pems03 | 0.1144±0.0001 | 0.3533±0.0002 | 0.0828±0.0000 | 0.0835±0.0001 | 0.0826±0.0001 | 0.0735±0.0000 |
| uber_hourly | N/A | N/A | 0.1488±0.0003 | 0.1468±0.0002 | 0.1576±0.0003 | 0.1762±0.0003 |
| | | | avg. rel. impr. | 3.59% | avg. rel. impr. | 3.13% |

Table 8: Comparison of 0.5-risk accuracy. "w/o" denotes methods without time-dependent errors, while "w/" indicates our method. Boldface values indicate that models considering time-dependent errors have better performance. Mean and standard deviation are obtained from 10 runs of each model.

| | VAR | GARCH | GPVar | | Transformer | |
|---|---|---|---|---|---|---|
| | | | w/o | w/ | w/o | w/ |
| exchange_rate | 0.0049±0.0000 | 0.0256±0.0001 | 0.0109±0.0003 | 0.0095±0.0004 | 0.0060±0.0001 | 0.0056±0.0001 |
| solar | 0.6140±0.0025 | 0.5621±0.0008 | 0.4998±0.0025 | 0.5246±0.0016 | 0.4233±0.0017 | 0.3958±0.0013 |
| electricity | 0.1113±0.0005 | 0.2014±0.0010 | 0.0405±0.0003 | 0.0397±0.0002 | 0.0449±0.0002 | 0.0505±0.0001 |
| traffic | 10.2654±0.0268 | 0.2722±0.0002 | 0.0933±0.0001 | 0.0859±0.0001 | 0.0803±0.0001 | 0.0794±0.0001 |
| wiki | 171.5009±0.2573 | 0.7225±0.0067 | 0.2231±0.0005 | 0.2236±0.0006 | 0.2030±0.0005 | 0.1487±0.0003 |
| m4_hourly | 0.1992±0.0003 | 0.2365±0.0005 | 0.0807±0.0001 | 0.0849±0.0001 | 0.0880±0.0001 | 0.0808±0.0001 |
| m1_quarterly | N/A | N/A | 0.2196±0.0023 | 0.1948±0.0005 | 0.2328±0.0008 | 0.2327±0.0014 |
| pems03 | 0.0784±0.0001 | 0.2028±0.0002 | 0.0568±0.0000 | 0.0574±0.0001 | 0.0569±0.0000 | 0.0506±0.0000 |
| uber_hourly | N/A | N/A | 0.1035±0.0002 | 0.1013±0.0002 | 0.1093±0.0003 | 0.1234±0.0002 |
| | | | avg. rel. impr. | 2.75% | avg. rel. impr. | 3.88% |

Table 9: Comparison of 0.9-risk accuracy. "w/o" denotes methods without time-dependent errors, while "w/" indicates our method. Boldface values indicate that models considering time-dependent errors have better performance. Mean and standard deviation are obtained from 10 runs of each model.

| | VAR | GARCH | GPVar | | Transformer | |
|---|---|---|---|---|---|---|
| | | | w/o | w/ | w/o | w/ |
| exchange_rate | 0.0021±0.0000 | 0.0070±0.0000 | 0.0042±0.0001 | 0.0057±0.0001 | 0.0030±0.0000 | 0.0023±0.0001 |
| solar | 0.4676±0.0016 | 0.4393±0.0008 | 0.1617±0.0004 | 0.1597±0.0003 | 0.2744±0.0015 | 0.2710±0.0015 |
| electricity | 0.0414±0.0003 | 0.0744±0.0003 | 0.0211±0.0004 | 0.0185±0.0002 | 0.0281±0.0002 | 0.0366±0.0002 |
| traffic | 11.0170±0.0405 | 0.1689±0.0001 | 0.0666±0.0001 | 0.0580±0.0001 | 0.0609±0.0001 | 0.0698±0.0000 |
| wiki | 174.0756±0.3770 | 1.5906±0.0044 | 0.2136±0.0002 | 0.2048±0.0001 | 0.2117±0.0006 | 0.1764±0.0003 |
| m4_hourly | 0.1029±0.0003 | 0.1309±0.0003 | 0.0452±0.0002 | 0.0463±0.0001 | 0.0525±0.0001 | 0.0475±0.0002 |
| m1_quarterly | N/A | N/A | 0.3049±0.0044 | 0.2787±0.0027 | 0.3784±0.0031 | 0.3621±0.0037 |
| pems03 | 0.0399±0.0000 | 0.1783±0.0001 | 0.0317±0.0000 | 0.0317±0.0001 | 0.0304±0.0000 | 0.0269±0.0000 |
| uber_hourly | N/A | N/A | 0.0533±0.0002 | 0.0528±0.0001 | 0.0562±0.0001 | 0.0638±0.0002 |
| | | | avg. rel. impr. | 0.22% | avg. rel. impr. | 0.91% |

Table 10: Comparison of ES accuracy. "w/o" denotes methods without time-dependent errors, while "w/" indicates our method. Boldface values indicate that models considering time-dependent errors have better performance. Mean and standard deviation are obtained from 10 runs of each model.

| | VAR | GARCH | GPVar | | Transformer | |
|---|---|---|---|---|---|---|
| | | | w/o | w/ | w/o | w/ |
| exchange_rate | 0.1301±0.0002 | 0.6085±0.0009 | 0.3674±0.0067 | 0.2613±0.0047 | 0.1798±0.0039 | 0.1438±0.0026 |
| solar $(\times 10^3)$ | 1.7429±0.0043 | 1.7758±0.0015 | 1.6052±0.0095 | 1.6591±0.0050 | 1.5307±0.0049 | 1.4633±0.0044 |
| electricity $(\times 10^5)$ | 1.0102±0.0052 | 1.9422±0.0127 | 0.3569±0.0050 | 0.3172±0.0028 | 0.4031±0.0040 | 0.4754±0.0025 |
| traffic | 3.3585±0.010 $(\times 10^3)$ | 4.4198±0.0020 | 2.4008±0.0020 | 2.2408±0.0015 | 2.2240±0.0021 | 2.2566±0.0014 |
| wiki $(\times 10^7)$ | 970.0242±2.5944 | 2.8857±0.0783 | 0.1149±0.0027 | 0.1155±0.0031 | 0.1236±0.004 | 0.1075±0.0046 |
| m4_hourly $(\times 10^3)$ | 4.5109±0.0084 | 5.1849±0.0089 | 2.2729±0.0062 | 2.3611±0.0060 | 2.5877±0.0098 | 2.3440±0.0081 |
| m1_quarterly $(\times 10^2)$ | N/A | N/A | 3.7565±0.0294 | 3.3676±0.0147 | 4.2149±0.0252 | 4.1596±0.0248 |
| pems03 $(\times 10^3)$ | 1.3951±0.0009 | 5.4642±0.0067 | 1.0535±0.0010 | 1.0736±0.0015 | 1.0673±0.0012 | 0.9394±0.0004 |
| uber_hourly $(\times 10^3)$ | N/A | N/A | 0.9035±0.0041 | 0.8773±0.0027 | 0.9377±0.0035 | 1.0566±0.0033 |
| | | | avg. rel. impr. | 5.58% | avg. rel. impr. | 3.21% |

Table 11: Comparison of RRMSE accuracy. "w/o" denotes methods without time-dependent errors, while "w/" indicates our method. Boldface values indicate that models considering time-dependent errors have better performance. Mean and standard deviation are obtained from 10 runs of each model.

| | VAR | GARCH | GPVar | | Transformer | |
|---|---|---|---|---|---|---|
| | | | w/o | w/ | w/o | w/ |
| exchange_rate | 0.0247±0.0000 | 0.0983±0.0002 | 0.0699±0.0012 | 0.0501±0.0010 | 0.0350±0.0008 | 0.0265±0.0007 |
| solar | 0.9365±0.0025 | 0.9556±0.0008 | 0.8195±0.0038 | 0.8334±0.0019 | 0.8114±0.0023 | 0.7761±0.0019 |
| electricity | 0.2732±0.0020 | 0.5584±0.0036 | 0.1010±0.0013 | 0.0912±0.0009 | 0.1130±0.0011 | 0.1293±0.0007 |
| traffic | 0.6312±0.0017 $(\times 10^3)$ | 0.9894±0.0008 | 0.5383±0.0005 | 0.5061±0.0003 | 0.5025±0.0005 | 0.5052±0.0003 |
| wiki | 0.6519±0.0016 $(\times 10^4)$ | 6.3386±0.2020 | 1.0288±0.0029 | 1.0393±0.0039 | 0.9292±0.0057 | 0.8752±0.0027 |
| m4_hourly | 0.6163±0.0012 | 0.6848±0.0015 | 0.3072±0.0008 | 0.3168±0.0007 | 0.3420±0.0011 | 0.3179±0.0010 |
| m1_quarterly | N/A | N/A | 19.1005±0.1246 | 17.0277±0.0845 | 20.2333±0.0830 | 20.2708±0.0924 |
| pems03 | 0.3727±0.0003 | 0.8824±0.0013 | 0.2796±0.0003 | 0.2877±0.0005 | 0.2841±0.0003 | 0.2502±0.0001 |
| uber_hourly | N/A | N/A | 0.2358±0.0012 | 0.2282±0.0008 | 0.2458±0.0010 | 0.2768±0.0009 |
| | | | avg. rel. impr. | 5.48% | avg. rel. impr. | 2.85% |

Table 12: Training cost comparison. "w/o" denotes methods without time-dependent errors, while "w/" indicates our method.

| | GPVar | | | | Transformer | | | |
|---|---|---|---|---|---|---|---|---|
| | w/o | | w/ | | w/o | | w/ | |
| | sec./epoch | epochs | sec./epoch | epochs | sec./epoch | epochs | sec./epoch | epochs |
| exchange_rate | 4.60 | 56 | 200.27 | 39 | 9.73 | 57 | 206.49 | 41 |
| solar | 6.18 | 39 | 74.16 | 51 | 15.37 | 121 | 181.26 | 132 |
| electricity | 7.44 | 71 | 119.06 | 94 | 19.38 | 63 | 103.15 | 65 |
| traffic | 10.50 | 55 | 225.20 | 48 | 28.30 | 84 | 247.58 | 100 |
| wiki | 12.45 | 30 | 164.81 | 33 | 28.16 | 51 | 351.06 | 48 |
| m4_hourly | 7.42 | 67 | 189.14 | 43 | 17.81 | 43 | 355.27 | 65 |
| m1_quarterly | 4.24 | 51 | 25.82 | 12 | 9.84 | 29 | 24.99 | 17 |
| pems03 | 11.75 | 62 | 143.34 | 57 | 36.31 | 78 | 88.05 | 53 |
| uber_hourly | 6.82 | 41 | 174.90 | 57 | 17.28 | 35 | 188.08 | 65 |

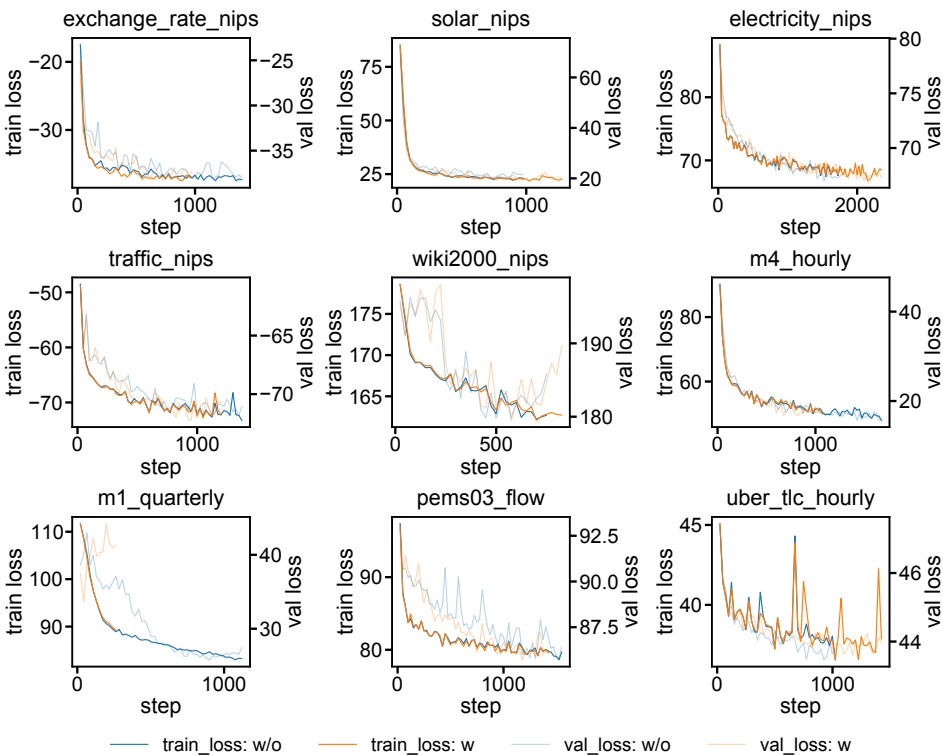

Figure 5: Training loss/validation loss vs training time of the GPVar model. "w/o" denotes methods without time-dependent errors, while "w/" indicates our method.

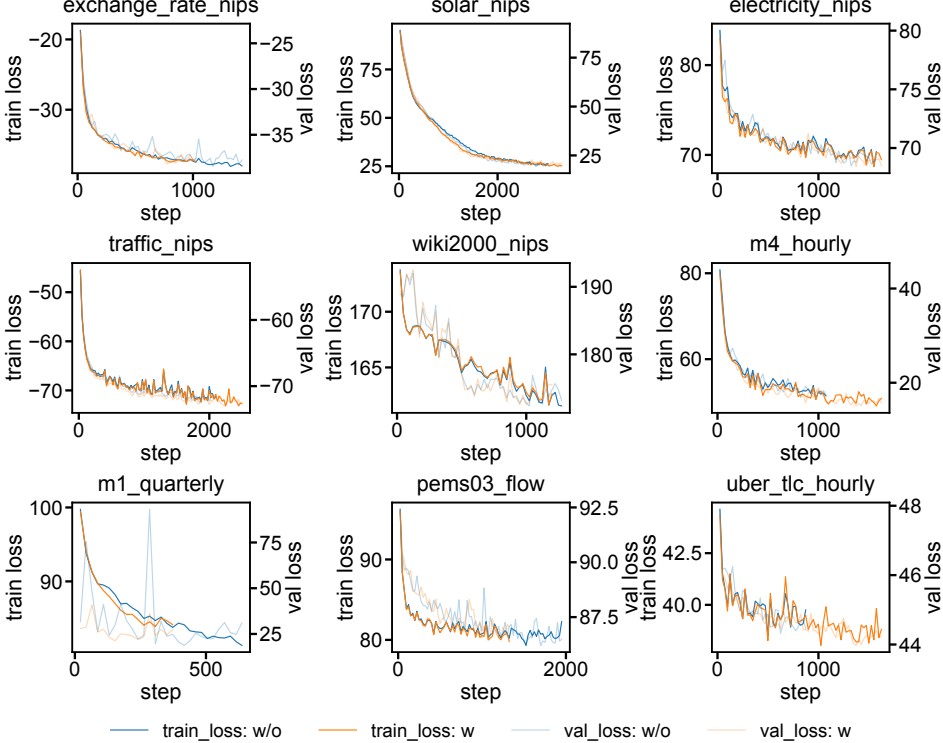

Figure 6: Training loss/validation loss vs training time of the Transformer model. "w/o" denotes methods without time-dependent errors, while "w/" indicates our method.

predictions. This allows more information to be utilized, potentially improving both predictions and error calibration, provided that memory capacity permits. We conducted an additional experiment to demonstrate the effect of increasing the batch size during inference (Fig. 7).

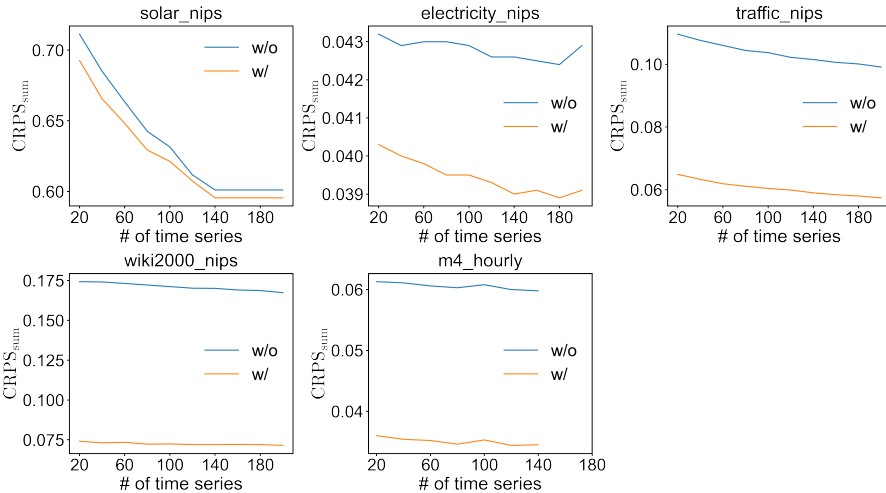

Figure 7: The influence of the number of time series in a batch on the performance of inference. "w/o" denotes methods without time-dependent errors, while "w/" indicates our method. We only show some datasets here because the remaining datasets have fewer than $B = 20$ time series in the testing set.

## B.5    Additional Model Interpretation

In this section, we provide further insights into how our method improves the base model. We illustrate these improvements by comparing the cross-correlations of the residuals from models with and without our method. Additionally, we demonstrate the performance of our method over the prediction horizon in multistep-ahead forecasting.

### B.5.1    Comparison of Residual Correlation

Recall that our method models both the autocovariance of errors $\text{Cov}(\eta_{i,t-\Delta}, \eta_{i,t})$ and the cross-lag covariance $\text{Cov}(\eta_{i,t-\Delta}, \eta_{j,t})$ between all pairs of components in the multivariate series. With the calibration process introduced in §4.2, our method is expected to reduce error cross-correlations, including autocorrelation and cross-lag correlation. Here, we compare the empirical ACF of the residuals $\eta_{i,t}$ of a single time series $i$, as well as the empirical cross-correlations of $\boldsymbol{\eta}_t$ across multiple time series.

We begin by comparing the ACF of the one-step-ahead prediction residuals with and without our method. The comparisons are provided for the following datasets: `solar` (Fig. 8), `electricity` (Fig. 9), `traffic` (Fig. 10), `wiki` (Fig. 11), `m4_hourly` (Fig. 12), `pems03` (Fig. 13), and `uber_hourly` (Fig. 14). We observe that the autocorrelation of the residuals is reduced after applying our method.

Next, we compare the cross-correlations of the one-step-ahead prediction residuals with and without our method. The comparisons are provided for the following datasets: `electricity` (Fig. 15), `traffic` (Fig. 16), `wiki` (Fig. 17), `m4_hourly` (Fig. 18), `pems03` (Fig. 19), and `uber_hourly` (Fig. 20). We also observe that the cross-correlations of the residuals are reduced after applying our method.

### B.5.2    Performance Breakdown at Each Forecast Step

To investigate our performance gain at each forecast step, we calculate the $\text{CRPS}_{\text{sum}}$ for each forecast step. The results are shown in Fig. 21 for GPVar and Fig. 22 for the Transformer. Note that the $\text{CRPS}_{\text{sum}}$ reported in this section may have different scales compared to previous sections because

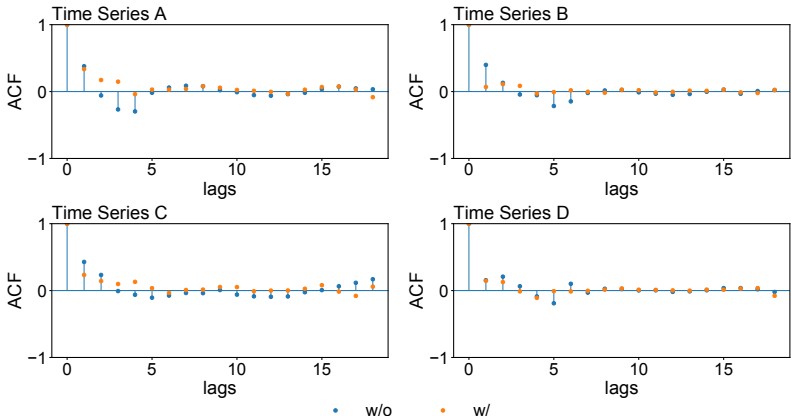

Figure 8: ACF comparison of the one-step-ahead prediction residuals with and without our method. The results depict the prediction outcomes generated by GPVar for four time series in the `solar` dataset.

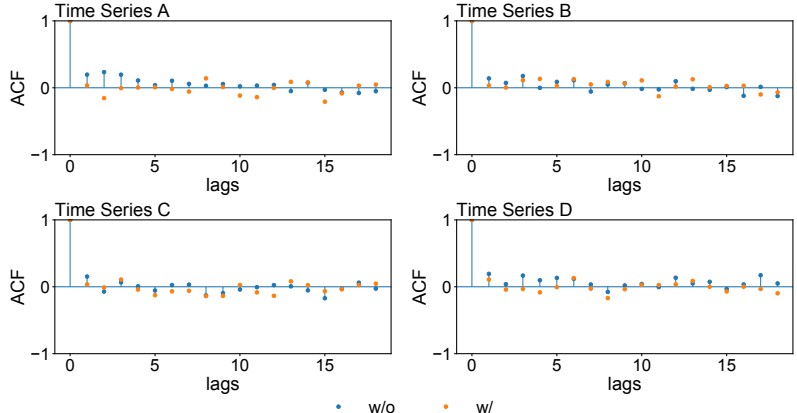

Figure 9: ACF comparison of the one-step-ahead prediction residuals with and without our method. The results depict the prediction outcomes generated by GPVar for four time series in the `electricity` dataset.

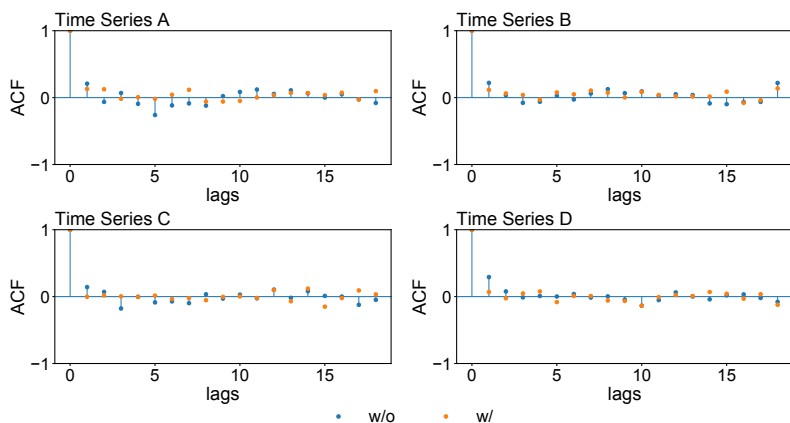

Figure 10: ACF comparison of the one-step-ahead prediction residuals with and without our method. The results depict the prediction outcomes generated by GPVar for four time series in the `traffic` dataset.

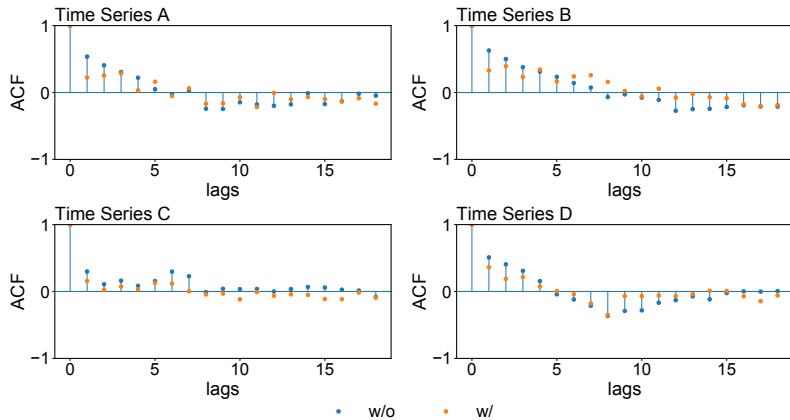

Figure 11: ACF comparison of the one-step-ahead prediction residuals with and without our method. The results depict the prediction outcomes generated by GPVar for four time series in the `wiki` dataset.

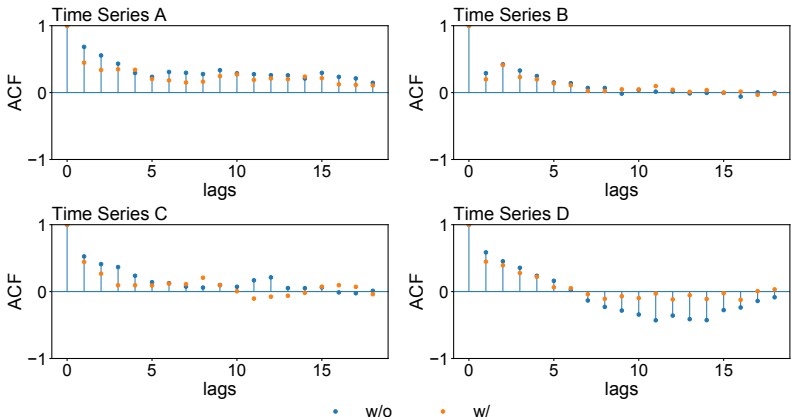

Figure 12: ACF comparison of the one-step-ahead prediction residuals with and without our method. The results depict the prediction outcomes generated by GPVar for four time series in the `m4_hourly` dataset.

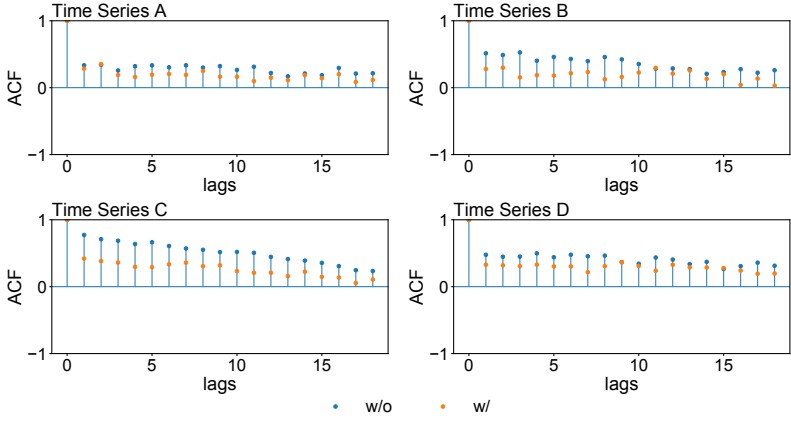

Figure 13: ACF comparison of the one-step-ahead prediction residuals with and without our method. The results depict the prediction outcomes generated by GPVar for four time series in the `pems03` dataset.

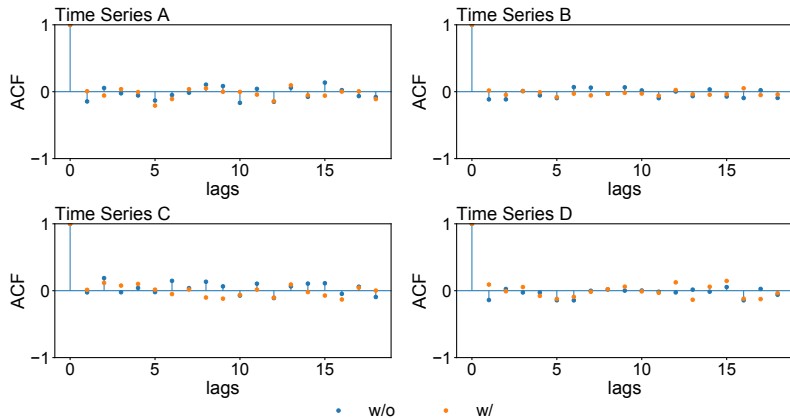

Figure 14: ACF comparison of the one-step-ahead prediction residuals with and without our method. The results depict the prediction outcomes generated by GPVar for four time series in the `uber_hourly` dataset.

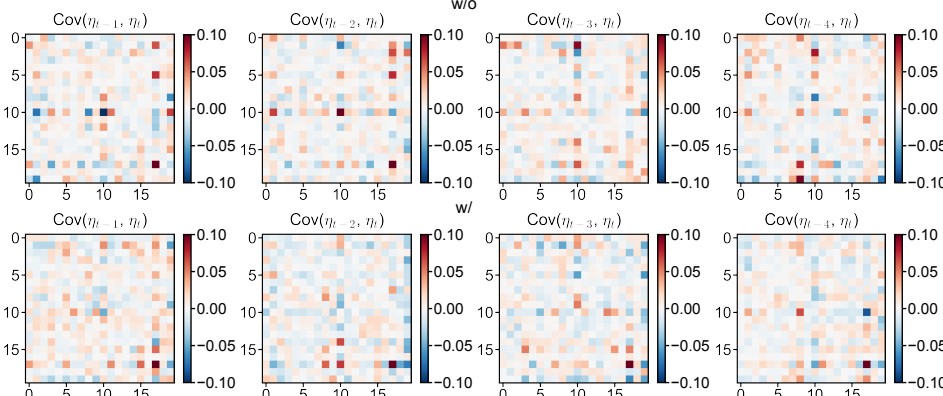

Figure 15: Cross-correlation comparison of the one-step-ahead prediction residuals with and without our method. The results depict the prediction outcomes generated by GPVar for four time series in the `electricity` dataset.

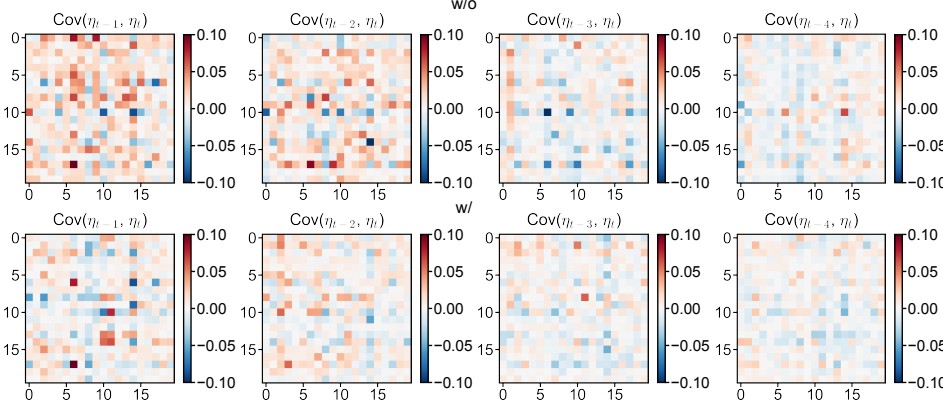

Figure 16: Cross-correlation comparison of the one-step-ahead prediction residuals with and without our method. The results depict the prediction outcomes generated by GPVar for four time series in the `traffic` dataset.

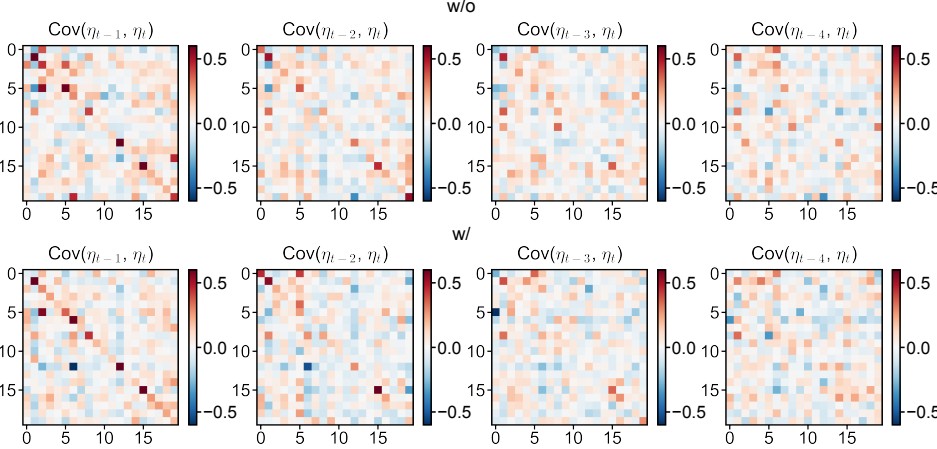

Figure 17: Cross-correlation comparison of the one-step-ahead prediction residuals with and without our method. The results depict the prediction outcomes generated by GPVar for four time series in the `wiki` dataset.

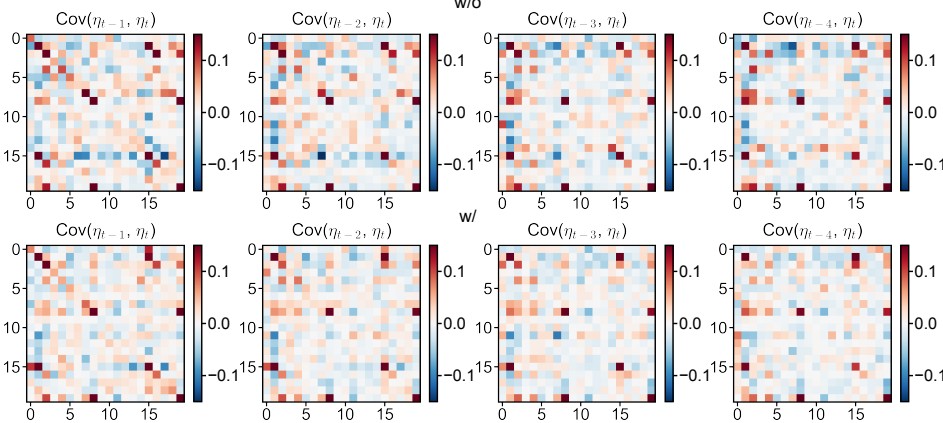

Figure 18: Cross-correlation comparison of the one-step-ahead prediction residuals with and without our method. The results depict the prediction outcomes generated by GPVar for four time series in the `m4_hourly` dataset.

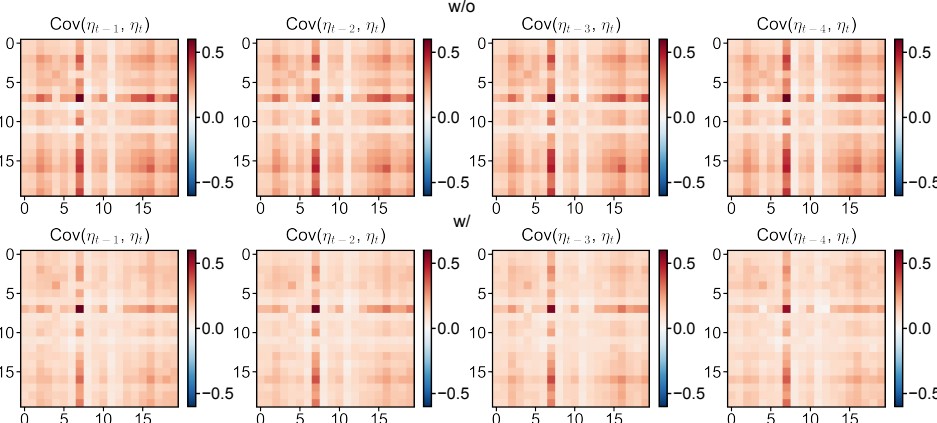

Figure 19: Cross-correlation comparison of the one-step-ahead prediction residuals with and without our method. The results depict the prediction outcomes generated by GPVar for four time series in the `pems03` dataset.

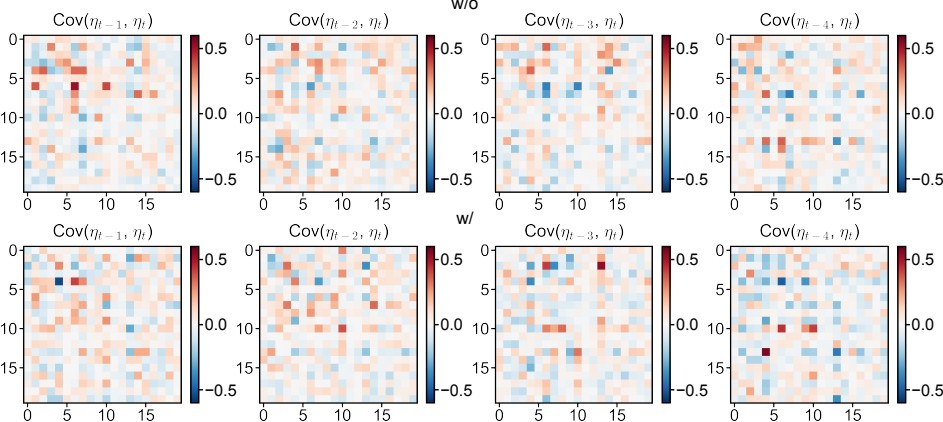

Figure 20: Cross-correlation comparison of the one-step-ahead prediction residuals with and without our method. The results depict the prediction outcomes generated by GPVar for four time series in the `uber_hourly` dataset.

they are not normalized. In multistep-ahead forecasting, since the predicted values are used as inputs for subsequent predictions within the prediction range, the residuals accumulate the effects of inaccuracies from previous steps. Therefore, the performance improvement depends not only on our modeling of error correlations but also on the properties of the residuals. These properties can be influenced by the absolute and relative time of the forecast and the seasonality of the data. For data without strong seasonality, residuals tend to be larger when predicting further ahead, making error accumulation more apparent. Conversely, for data with strong seasonality, the impact of error accumulation can vary. We observe that, in most scenarios, $\mathrm{CRPS_{sum}}$ is reduced at the early forecasting stages. As predictions extend further into the future, some datasets (e.g., `traffic` in Fig. 21) show decreased improvement, likely due to seasonality effects. Conversely, other datasets (e.g., `wiki` in Fig. 21) exhibit larger improvements further into the future, possibly because the residuals accumulate over the steps.

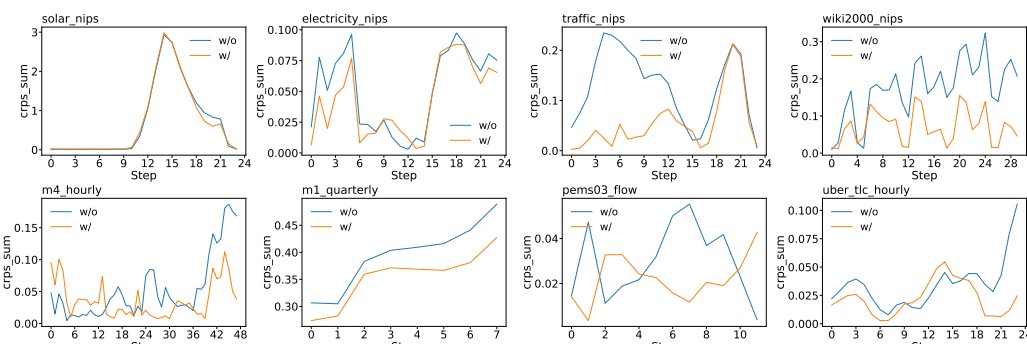

Figure 21: Step-wise $\mathrm{CRPS_{sum}}$ accuracy of GPVar. "w/o" denotes methods without time-dependent errors, while "w/" indicates our method.

## B.6 Alternative Parametrization of $C_t$

### B.6.1 Learnable Lengthscales

In this paper, the lengthscales are fixed when generating the correlation matrix $C_t$, and the flexibility of $C_t$ comes from dynamically generating the component weights of the kernel matrices. Making these lengthscales learnable parameters to find the optimal set of $\{l_m\}_{m=1}^{M-1}$ is another approach we can explore to increase modeling flexibility. Based on the best model identified in Table 1, we experiment with treating the lengthscales as learnable parameters, jointly optimized with the base

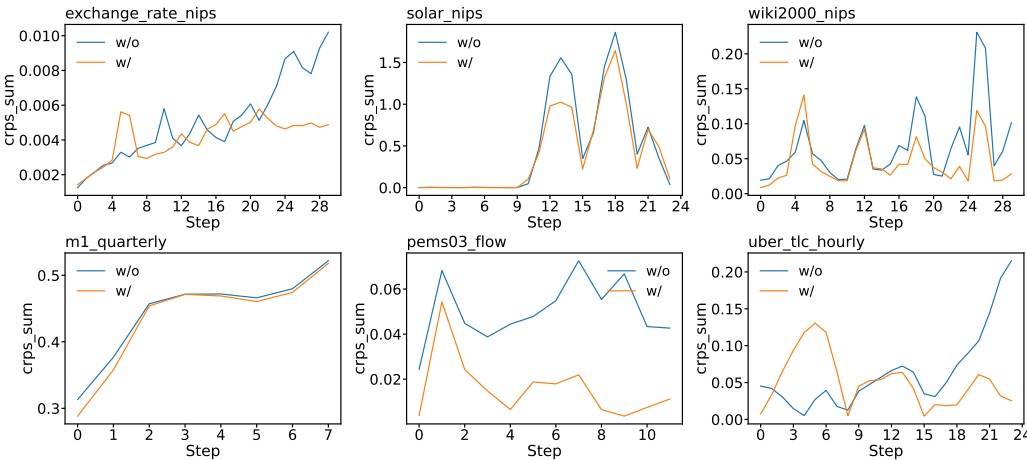

Figure 22: Step-wise $\text{CRPS}_{\text{sum}}$ accuracy of Transformer. "w/o" denotes methods without time-dependent errors, while "w/" indicates our method.

model. The results are shown in Table 13. We do not observe significant improvement from learnable lengthscales.

Table 13: Comparison of $\text{CRPS}_{\text{sum}}$ accuracy. "w/o" denotes methods without time-dependent errors, while "w/" indicates our method. "w/ ($l$)" indicates the lengthscales are learnable parameters. Boldface values indicate that models considering time-dependent errors have better performance. Mean and standard deviation are obtained from 10 runs of each model.

| | GPVar | | | Transformer | | |
|---|---|---|---|---|---|---|
| | w/o | w/ | w/ ($l$) | w/o | w/ | w/ ($l$) |
| exchange_rate | 0.0068±0.0004 | 0.0117±0.0004 | **0.0045±0.0001** | 0.0055±0.0002 | **0.0042±0.0002** | 0.0072±0.0002 |
| solar | 0.7103±0.0065 | **0.6929±0.0039** | 0.7727±0.0040 | 0.4960±0.0034 | **0.4132±0.0027** | 0.4138±0.0023 |
| electricity | 0.0430±0.0005 | 0.0403±0.0004 | **0.0351±0.0003** | 0.0638±0.0003 | 0.0638±0.0003 | 0.0858±0.0006 |
| traffic | 0.1095±0.0002 | **0.0649±0.0002** | 0.1297±0.0003 | **0.0717±0.0002** | 0.0981±0.0002 | 0.0950±0.0002 |
| wiki | 0.1745±0.0008 | **0.0743±0.0009** | 0.4839±0.0021 | 0.0841±0.0013 | 0.0500±0.0005 | **0.0472±0.0004** |
| m4_hourly | 0.0613±0.0004 | **0.0358±0.0002** | 0.0527±0.0003 | 0.0651±0.0004 | 0.0616±0.0003 | **0.0355±0.0003** |
| m1_quarterly | 0.3942±0.0030 | 0.3538±0.0017 | **0.3534±0.0017** | 0.4448±0.0027 | 0.4367±0.0028 | **0.3709±0.0120** |
| pems03 | 0.0503±0.0001 | 0.0491±0.0002 | **0.0456±0.0001** | 0.0490±0.0001 | 0.0386±0.0001 | **0.0330±0.0001** |
| uber_hourly | 0.0342±0.0006 | 0.0222±0.0004 | **0.0218±0.0003** | 0.0632±0.0003 | **0.0513±0.0005** | 0.0969±0.0005 |

### B.6.2 Using Autocorrelations of an AR($p$) process

One could parameterize $C_t$ as fully learnable, positive definite symmetric Toeplitz matrices. For instance, an AR($p$) process has an autocorrelation matrix with a Toeplitz structure, allowing the modeling of negative correlations. This alternative approach may offer more flexibility in capturing complex correlation patterns in multivariate time series data. The autocorrelations of an AR($p$) process can be obtained by solving a set of equations known as the Yule-Walker equations [46]. For example, if we consider an AR(2) process and let $\rho_k$ be the autocorrelation at lag $k$:

$$z_t = \phi_1 z_{t-1} + \phi_2 z_{t-2} + \epsilon_t. \tag{32}$$

where $\phi_1$ and $\phi_2$ are the coefficients. We have $\rho_0 = 1$ by definition and:

$$\begin{aligned} \rho_1 &= \phi_1 \rho_0 + \phi_2 \rho_1, \\ &\cdots \\ \rho_k &= \phi_1 \rho_{k-1} + \phi_2 \rho_{k-2}, k \geq 2. \end{aligned} \tag{33}$$

Since $\rho_0 = 1$, we can solve for $\rho_1$:

$$\rho_1 = \frac{\phi_1}{1 - \phi_2}, \tag{34}$$

and for any $k \geq 2$, we can solve $\rho_k$ iteratively by:

$$\rho_k = \phi_1 \rho_{k-1} + \phi_2 \rho_{k-2}, k \geq 2. \tag{35}$$

The collection $\{\rho_0, \rho_1, \ldots, \rho_k, \ldots, \rho_{D-1}\}$ forms the first row or column of a Toeplitz matrix and can be used to parameterize $C_t$. We perform a hyperparameter search to find the best AR order $p$ based on the validation loss. As shown in Table 14, while the correlation matrix $C_t$ parameterized by an AR process shows promise in modeling both positive and negative correlations, it does not empirically provide an overall improvement compared to the kernel method used in this paper. This may be because cross-correlations in time series are predominantly positive. However, the AR method does show significant improvements on certain datasets where the kernel method does not perform well. For example, the AR method greatly improves GPVar on `exchange_rate` and the Transformer on `electricity`.

Table 14: Comparison of $\text{CRPS}_{\text{sum}}$ accuracy. "w/o" denotes methods without time-dependent errors, while "w/" indicates our method. "w/ (AR)" indicates $C_t$ is parameterized by an AR process. Boldface values indicate that models considering time-dependent errors have better performance. Mean and standard deviation are obtained from 10 runs of each model.

| | GPVar | | | Transformer | | |
|---|---|---|---|---|---|---|
| | w/o | w/ | w/ (AR) | w/o | w/ | w/ (AR) |
| exchange_rate | 0.0068±0.0004 | 0.0117±0.0004 | **0.0051±0.0002** | 0.0055±0.0002 | **0.0042±0.0002** | 0.0088±0.0004 |
| solar | 0.7103±0.0065 | 0.6929±0.0039 | **0.5923±0.0042** | 0.4960±0.0034 | 0.4132±0.0027 | **0.3362±0.0025** |
| electricity | 0.0430±0.0005 | **0.0403±0.0004** | 0.0433±0.0007 | 0.0494±0.0004 | 0.0638±0.0003 | **0.0252±0.0002** |
| traffic | 0.1095±0.0002 | **0.0649±0.0002** | 0.1095±0.0004 | **0.0717±0.0002** | 0.0981±0.0002 | 0.0878±0.0003 |
| wiki | 0.1745±0.0008 | **0.0743±0.0009** | 0.2375±0.0013 | 0.0841±0.0013 | **0.0500±0.0005** | 0.0512±0.0008 |
| m4_hourly | 0.0613±0.0004 | 0.0358±0.0002 | **0.0298±0.0002** | 0.0651±0.0004 | **0.0616±0.0003** | 0.0680±0.0003 |
| m1_quarterly | 0.3942±0.0030 | 0.3538±0.0017 | **0.1692±0.0029** | 0.4448±0.0027 | 0.4367±0.0028 | **0.4348±0.0028** |
| pems03 | 0.0503±0.0001 | **0.0491±0.0002** | 0.0787±0.0002 | 0.0490±0.0001 | **0.0386±0.0001** | 0.0656±0.0001 |
| uber_hourly | 0.0342±0.0006 | **0.0222±0.0004** | 0.0375±0.0004 | 0.0632±0.0003 | **0.0513±0.0005** | 0.0770±0.0007 |

## B.7 Alternative Error Assumptions

A more suitable likelihood function can regularize the training process, potentially reducing residual correlations. For example, assuming the errors follow a multivariate $t$-distribution improves the robustness of the model to outliers. Additionally, a stronger base model can help produce residuals that are more independent. Based on these considerations, we designed our approach to adapt dynamically to varying levels of error correlation. The weighted correlation matrix assigns greater weight to the identity matrix when the errors exhibit lower correlation.

We also trained the baseline models using the likelihood of the multivariate $t$-distribution, and the results are shown in Table 15. While using an alternative distribution can lead to better performance on certain datasets when our method is not applied, we observed that our method effectively closes the performance gap in cases where the multivariate Gaussian assumption is outperformed by the $t$-distribution.

An important feature of our method is the ability to use a subset of time series in each training batch for model optimization, which enhances scalability. For the multivariate $t$-distribution, the distribution of these subsets of $\mathbf{z}_t$ should have the same degrees of freedom as the full distribution of $\mathbf{z}_t$. However, since the degrees of freedom are treated as an additional output of the model in each training batch, they are not guaranteed to be consistent across batches. While this is not problematic for deep learning, it violates the marginalization property of the $t$-distribution from a statistical standpoint.

We chose Gaussian noise for its beneficial properties, including its marginalization rule and well-defined conditional distribution, both essential for statistically consistent model training and reliable inference. To address model misspecification, a more effective approach could involve first transforming the original observations into Gaussian-distributed data using a Gaussian Copula [3], and then applying our method.

Table 15: $\text{CRPS}_{\text{sum}}$ accuracy comparison. "w/o" denotes methods without time-dependent errors, while "w/" indicates our method. Bold values show models with time-dependent errors performing better. Mean and standard deviation are obtained from 10 runs of each model. "N/A" indicates that the model could not be properly fitted..

| | GPVar | | | Transformer | | |
|---|---|---|---|---|---|---|
| | Gaussian (w/o) | Gaussian (w/) | $t$-distribution (w/o) | Gaussian (w/o) | Gaussian (w/) | $t$-distribution (w/o) |
| exchange_rate | **0.0068±0.0004** | 0.0117±0.0004 | 0.0159±0.0005 | 0.0055±0.0002 | **0.0042±0.0002** | 0.0101±0.0003 |
| solar | 0.7103±0.0065 | **0.6929±0.0039** | N/A | 0.4960±0.0034 | **0.4132±0.0027** | N/A |
| electricity | 0.0430±0.0005 | **0.0403±0.0004** | 0.0467±0.0004 | 0.0494±0.0004 | 0.0638±0.0003 | **0.0466±0.0002** |
| traffic | 0.1095±0.0002 | **0.0649±0.0002** | 0.0679±0.0002 | **0.0717±0.0002** | 0.0981±0.0002 | N/A |
| wikipedia | 0.1745±0.0008 | 0.0743±0.0009 | **0.0730±0.0004** | 0.0841±0.0013 | **0.0500±0.0005** | 0.1979±0.0005 |
| m4_hourly | 0.0613±0.0004 | **0.0358±0.0002** | 0.0365±0.0003 | 0.0651±0.0004 | **0.0616±0.0003** | 0.0665±0.0003 |
| m1_quarterly | 0.3942±0.0030 | **0.3538±0.0017** | 0.3550±0.0084 | 0.4448±0.0027 | **0.4367±0.0028** | 0.4466±0.0044 |
| pems03 | 0.0503±0.0001 | **0.0491±0.0002** | 0.0679±0.0002 | 0.0490±0.0001 | **0.0386±0.0001** | 0.0529±0.0002 |
| uber_hourly | 0.0342±0.0006 | **0.0222±0.0004** | 0.0666±0.0010 | 0.0632±0.0003 | 0.0513±0.0005 | **0.0340±0.0004** |

## B.8 Qualitative Results on Forecasting

In this section, we provide qualitative analysis of the actual prediction performance by visualizing the predictions.

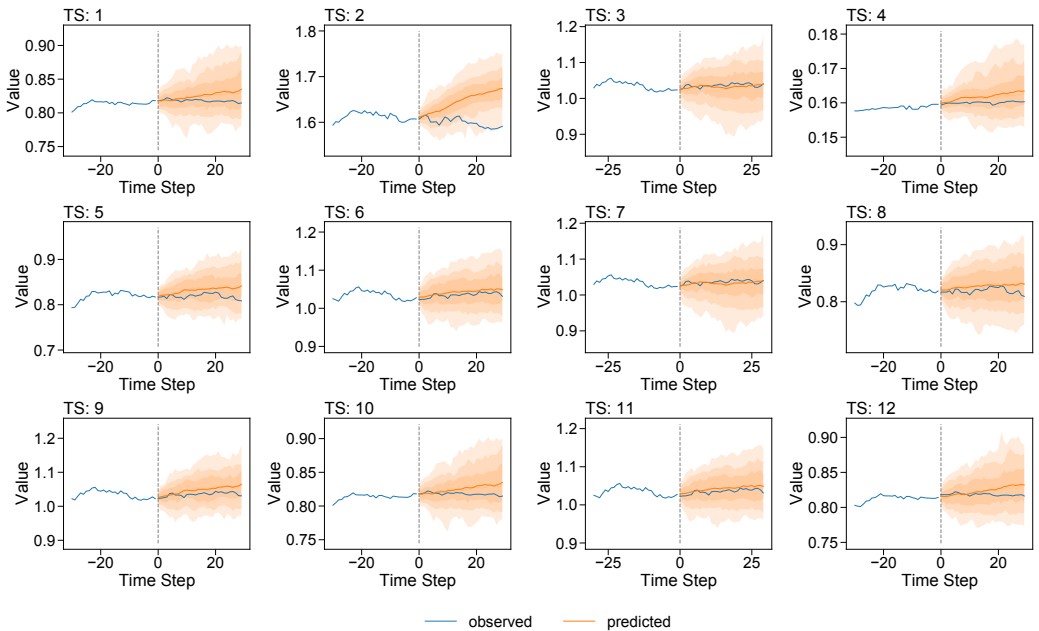

Figure 23: Visualization of forecasting results on exchange_rate using GPVar with our method.

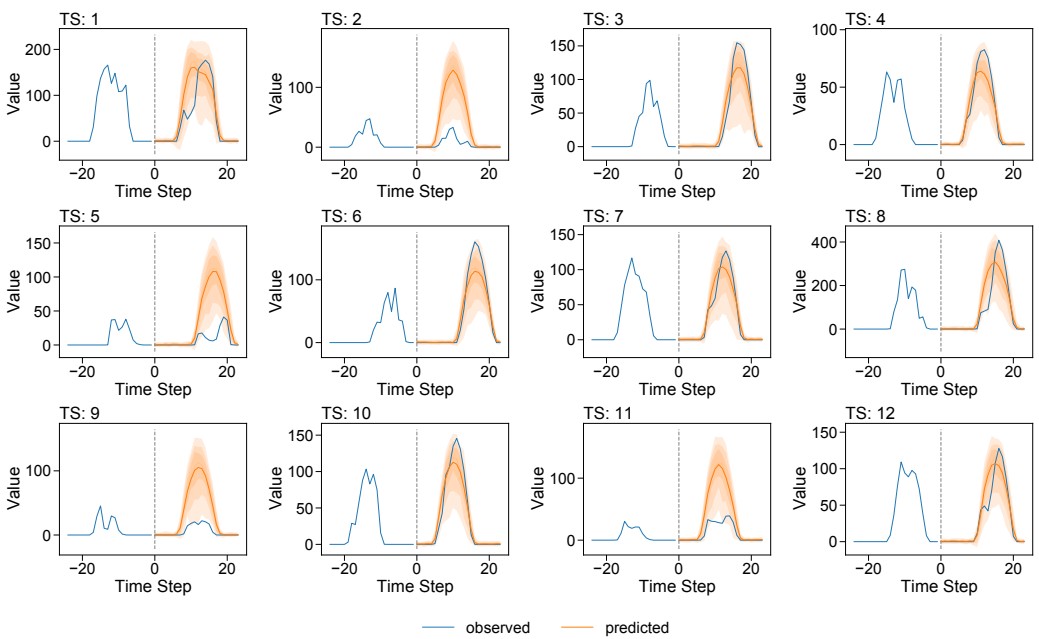

Figure 24: Visualization of forecasting results on `solar` using GPVar with our method.

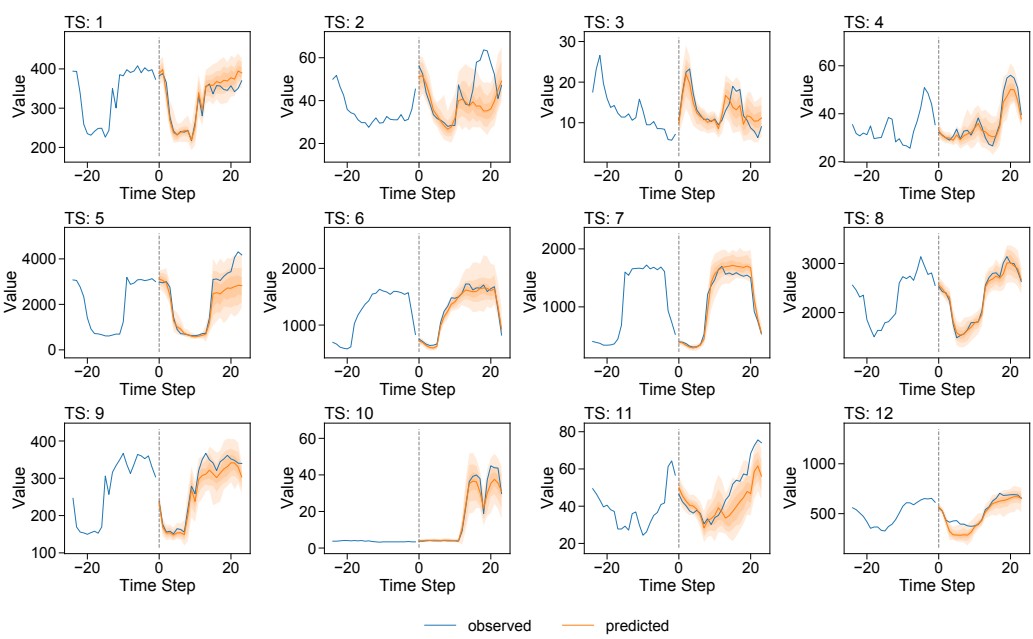

Figure 25: Visualization of forecasting results on `electricity` using GPVar with our method.

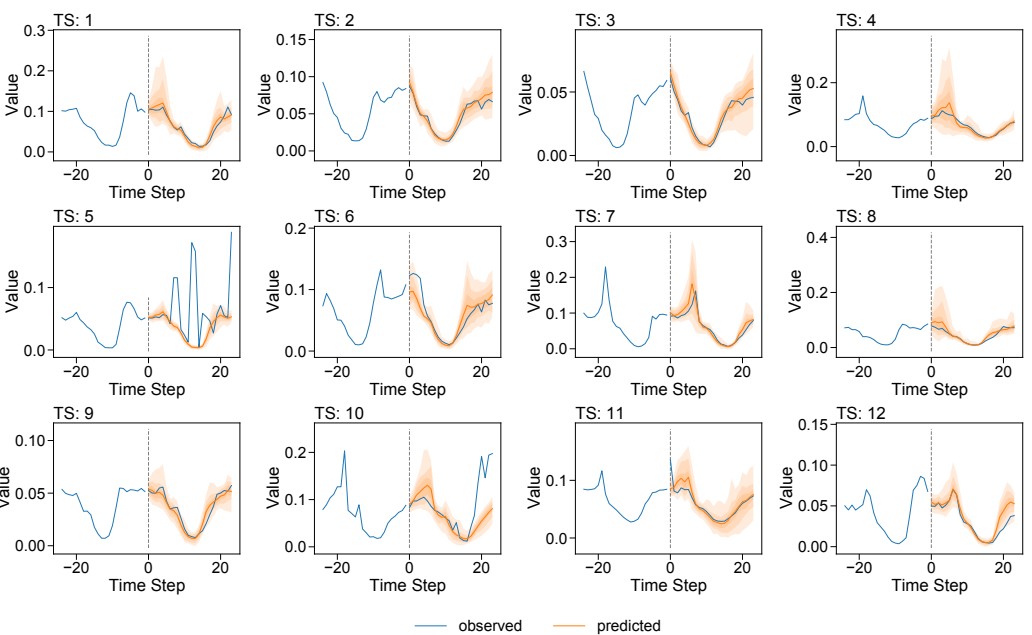

Figure 26: Visualization of forecasting results on `traffic` using GPVar with our method.

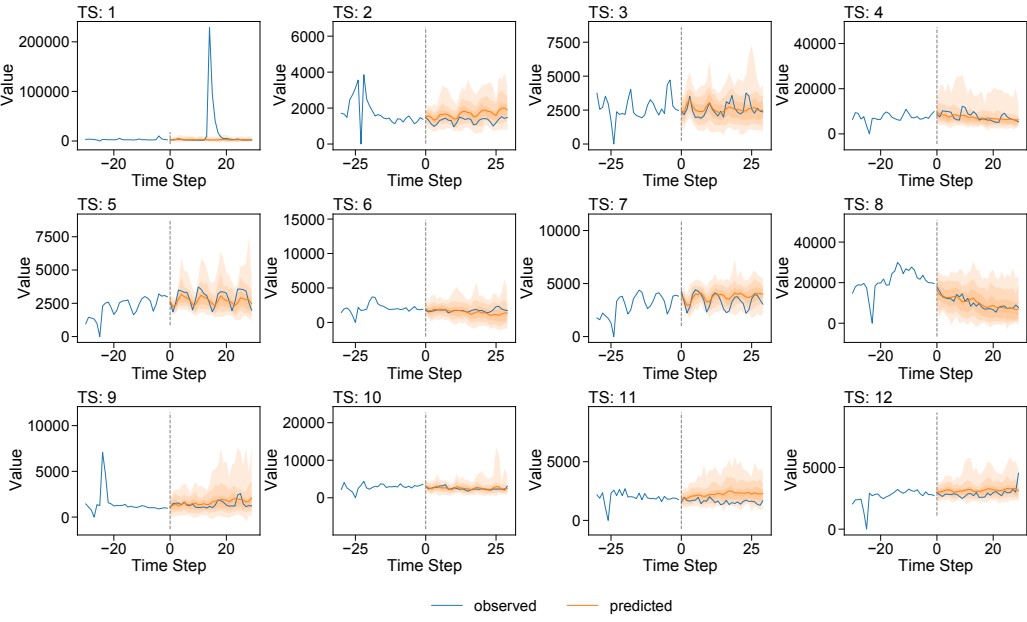

Figure 27: Visualization of forecasting results on `wiki` using GPVar with our method.

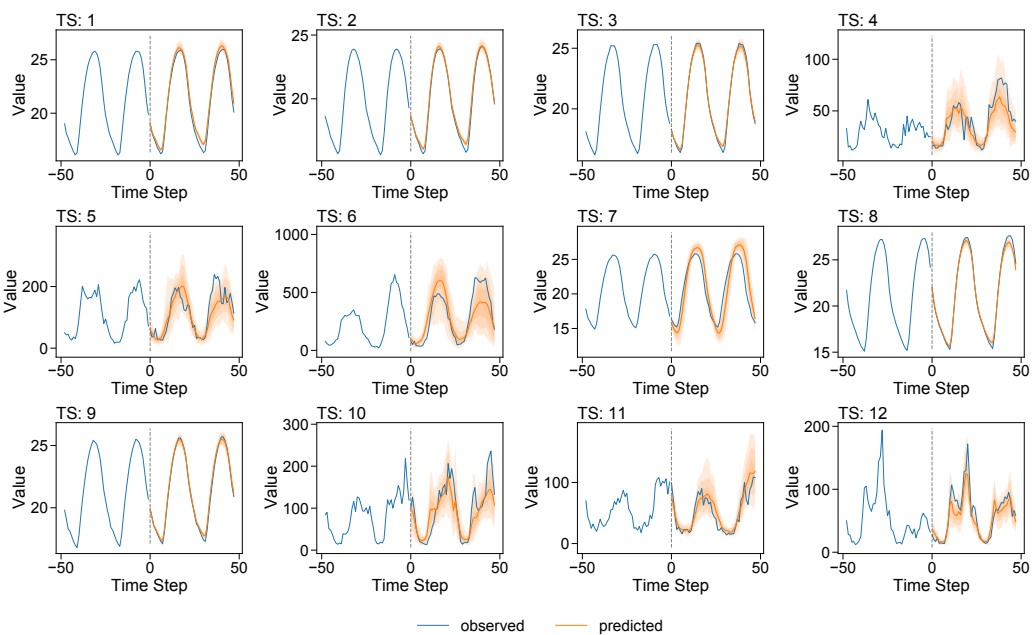

Figure 28: Visualization of forecasting results on `m4_hourly` using GPVar with our method.

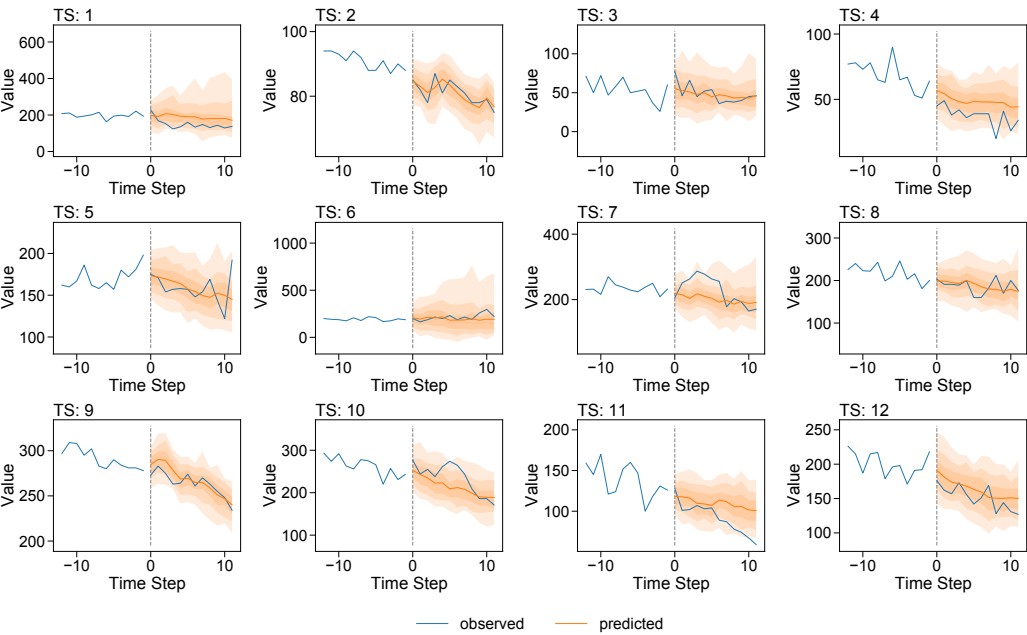

Figure 29: Visualization of forecasting results on `pems03` using GPVar with our method.

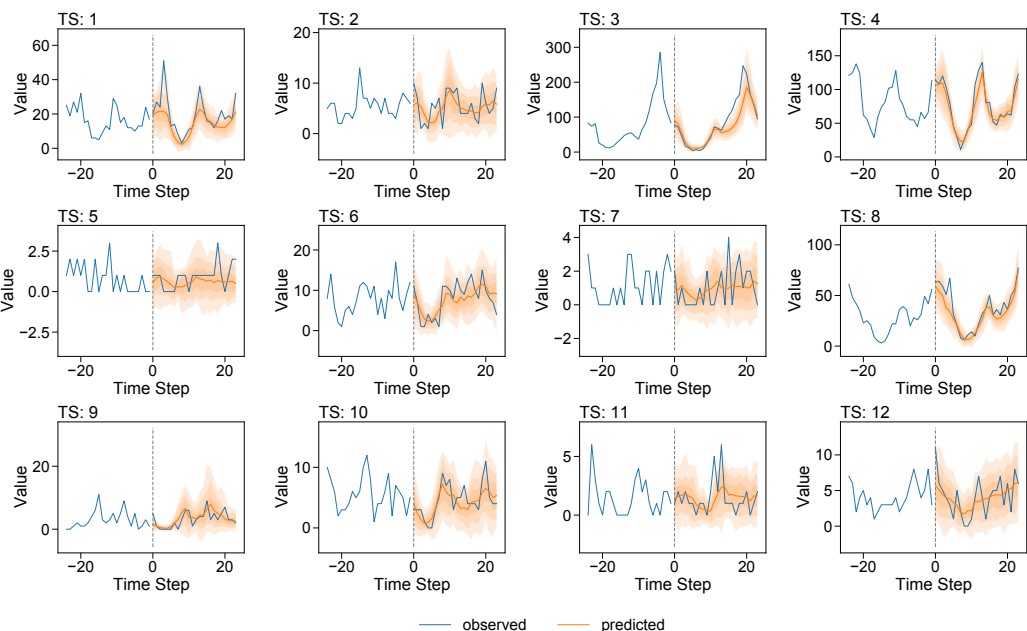

Figure 30: Visualization of forecasting results on `uber_hourly` using GPVar with our method.

