# OpenReview forum: "Multivariate Probabilistic Time Series Forecasting with Correlated Errors"
_NeurIPS.cc/2024/Conference — NeurIPS 2024 poster_

### Official Review · Reviewer_szpP · 2024-06-20

**Soundness:** 3
**Presentation:** 4
**Contribution:** 3
**Rating:** 7
**Confidence:** 4

**Summary:**

Briefly summarise the paper and its contributions. This is not the place to critique the paper; the authors should generally agree with a well-written summary.

The paper proposes an extension to multivariate forecasting model that takes into account the error auto-correlation. The issue of remaining auto-correlation residual is first illustrated clearly in an example with a multivariate forecasting method and the author then introduces their method. The method consists in modelling the cross-covariance through a learned weighted sum of kernels matrices that are projected to obtain the final estimated correlation. This approach leverages previous work on low-rank parametrisation of multivariate forecasting and the authors manages to find an approach that remains tractable despite the complexity of the task. Experiments are conducted on various real-world datasets where their method is shown to improve several forecasting metrics such as CRPS-SUM, RMSE or energy scores.

**Strengths:**

* the paper is well written, even though the approach is very technically involved, the authors did a great job at explaining it well
* the paper is well motivated: insighful data-analysis are made to illustrate the presence of autocorrelated residual and motivate their approach
* large coverage of experiments and empirical evaluations: the experiments are done on a large set of datasets covering many metrics and are also well assessed qualitively

**Weaknesses:**

* potential gaps in experiments:
  * matching budget: the method is currently much more expensive that the baseline considered
  * some details of Hyperparameter Optimization are currently missing (but will be added at the camera ready according to the rebuttal)
* the paper lacks clarity on whether the method will be competitive with better likelihoods. Currently the method may work well only because of non-competitive likelihood is used (Gaussian) as opposed to Student-t or Non-paranormal which would get better residuals

**Questions:**

I am separating my questions between the most important ones that would help to raise my score and minor comments.

## Most important

**Lack of clarity whether the method will be competitive with better likelihoods.** Currently, the method only studies a Gaussian noise. However, this noise is a poor fit for the dataset studied and as such is inflating the residuals. This leaves open the question of the method works just because the initial residuals are very poor (and adds a smoothing effect which helps their estimation) and not so much because of the reason given by the paper (that the approach models the autocorrelation aspect).

I can see two ways to remove this potential cofounding factor, one is to include experiments with paranormal distribution which would be more robust from [2], the other is to consider student-t distribution which should also be tractable.

**Details of Hyperparameter Optimization.** The authors mentions:
```
The best hyperparameters for a base model—dataset combination are selected based on the best validation loss. Once the optimal learning rate and hidden size are determined, we apply the same hyperparameters for models both with and without applying our approach
```
How does it work exactly? Which model are you using to determine and fix the hyperparameters? (Selecting the best hyperparameters for your method may favour their transformation)

**Matching budget.** The method proposed by the author is roughly 10x slower on average (and up to 30x slower for exchange-rate from table 10). This point is important and currently a bit buried down, for instance the authors mentions: “* However, there is no evidence that our method slows down convergence. On the contrary, it accelerates convergence for 5 out of 9 datasets for GPVar.” However, the wallclock time are clearly much slower I think this is fine for the paper overall but it should be stated clearly as a limitation (the method proposed by the authors would likely be worse than 10 ensemble of the baseline for instance, I don’t expect this particular point to appear in text or experiment but I want to explain why this is important).

**Citation of multivariate evaluation metrics.** The authors uses CRPS-sum as their main error metrics (in addition to many other metrics including energy scores in their appendix), I would highly recommend citing this paper which discuss the limitation for this metric when assessing multivariate methods (not mine):

*Regions of Reliability in the Evaluation of Multivariate Probabilistic Forecasts*
Étienne Marcotte, Valentina Zantedeschi, Alexandre Drouin, Nicolas Chapados
*Proceedings of the 40th International Conference on Machine Learning*, PMLR 202:23958-24004, 2023.

To be clear, I do not think this is a large problem given that the authors also conducted many high quality qualitative analysis of their results.

## Details
- It would be very useful to clearly spell out the dimension/numbers of projection parameters in a table, possibly in the appendix
- I highly encourage the authors to release their code (the intention is mentioned in the supp, the work will be hard to reproduce w/o given that it is quite technical, and the work will be more impactful as other will be able to directly compare to their setup)
- a small point regarding "real-world data frequently exhibit significant error autocorrelation and cross-lag correlation due to factors such as missing covariates", I agree that missing covariates is indeed a potential use-case but it feels very niche to me (none of the dataset have covariates for instance) compared to model mispecification

**Limitations:**

Yes

---

> ### Author Rebuttal · Authors · 2024-08-05
>
> We sincerely appreciate the reviewer's encouraging comments and insightful suggestions.
>
> # Questions
> ## Most important
> > Lack of clarity whether the method will be competitive with better likelihoods.
>
> ***Response:*** Thank you for your valuable feedback. The reviewer is correct in noting that a better likelihood function could potentially regularize the training process, leading to less correlated residuals. For instance, assuming the errors follow a multivariate $t$-distribution can enhance the model's robustness to outliers. Additionally, a more powerful base model can also produce more independent residuals. Given these factors, we designed our approach to dynamically adapt to varying levels of error correlation. Our weighted correlation matrix can assign a higher weight to the identity matrix when errors are less correlated.
>
> **We tested training the baseline models using the likelihood of the multivariate $t$-distribution, and the results are presented in Table 1 (see PDF for the global rebuttal).** It is true that using an alternative distribution can achieve better performance in certain datasets when our method is not implemented. However, we also observed that our method successfully remedies the performance gap for those datasets where the Gaussian assumption is outperformed by the $t$-distribution.
>
> *Note: An important aspect of our method is the ability to use a subset of time series in a training batch for model optimization, which improves scalability. For the multivariate $t$-distribution, the distribution of the subsets of $\boldsymbol{z}\_t$ should have the same degrees of freedom as the distribution of $\boldsymbol{z}\_t$. Since the degrees of freedom becomes an additional output of the model in each training batch, they are not guaranteed to be the same. While this is not a problem for deep learning, it violates the marginalization rule of the $t$-distribution from a statistical perspective.*
>
> Therefore, we chose Gaussian noise for its numerous beneficial properties, including its marginalization rule and well-defined conditional distribution, which are essential for statistically consistent model training and reliable inference. To address model misspecification, a more effective approach could involve first transforming the original observations $\mathbf{z}\_{t}$ into Gaussian-distributed data $\mathbf{x}\_{t}$ using a Gaussian Copula [1], and then applying our method.
>
> > Details of Hyperparameter Optimization.
>
> ***Response:*** Thank you for bringing this to our attention. When optimizing the hyperparameters of our model, we first tune each base model (e.g., GPVar) on each dataset individually and select the optimal hyperparameters based on the loss observed on the validation set. For example, there will be an optimal learning rate and hidden size for the GPVar model on the $\mathtt{traffic}$ dataset. We perform this tuning using the baseline models (i.e., without implementing our method). To ensure fair comparison, we use the same set of hyperparameters for the models implemented with our method.
>
> > Matching budget.
>
> ***Response:*** The reviewer rightly points out the increased training cost introduced by our method. In Table 10, the clock time per epoch for training with our method is significantly higher compared to the baseline methods. This increase is attributed to the larger $DB \times DB$ covariance matrix used in the likelihood calculation. We will clearly state this limitation in the next version of the paper.
>
> > Citation of multivariate evaluation metrics.
>
> ***Response:*** The reviewer is correct about the limitations of $\operatorname{CRPS}\_{\text{sum}}$ in assessing multivariate probabilistic forecasting, despite its wide usage in existing works [1-5]. $\operatorname{CRPS}\_{\text{sum}}$ has been reported to overlook the performance of the model on each dimension [6]. In [7], the authors suggest that neither $\operatorname{CRPS}$ nor the energy score is a good surrogate for the NLL score in small-sample regimes, which are common in practice. However, comparing NLL is infeasible in our case, as sampling must be conducted to calibrate predictions autoregressively over the prediction range. Therefore, we include various metrics for assessment in this paper. We will discuss the limitations of $\operatorname{CRPS}\_{\text{sum}}$ in the next version of the paper.
>
> [1] Salinas, David, et al. High-dimensional multivariate forecasting with low-rank gaussian copula processes. In NeurIPS, 2019.
>
> [2] Rasul, Kashif, et al. Multivariate probabilistic time series forecasting via conditioned normalizing flows. In ICLR, 2021.
>
> [3] Rasul, Kashif, et al. Autoregressive denoising diffusion models for multivariate probabilistic time series forecasting. In ICML, 2021.
>
> [4] Drouin, Alexandre, et al. Tactis. In ICML, 2022.
>
> [5] Ashok, Arjun, et al. Tactis-2. In ICLR, 2024.
>
> [6] Koochali, Alireza, et al. Random noise vs. state-of-the-art probabilistic forecasting methods: A case study on CRPS-Sum discrimination ability. Applied Sciences 12.10 (2022): 5104.
>
> [7] Marcotte, Étienne, et al. Regions of reliability in the evaluation of multivariate probabilistic forecasts. In ICML, 2023.
>
> ## Details
> > It would be very useful ..
>
> ***Response:*** Thank you for the suggestion. We have provided more details about the number of projection parameters (see Table 2 and Table 3 in the PDF for the global rebuttal.). It can be seen that we introduce only a negligible amount of new parameters to the model.
>
> > I highly encourage ...
>
> ***Response:*** Yes, we will release the code as soon as the work is published.
>
> > a small point regarding ...
>
> ***Response:*** We indeed have covariates for each dataset, which include generic features that encode time and time series identification, as detailed in Appendix A.6. We agree that a better likelihood, a stronger base forecasting model, and comprehensive covariates all contribute to less correlated residuals.

---

> ### Comment · Reviewer_szpP · 2024-08-10
> **Answer to rebuttal**
>
> Thank you for your very clear answer which addressed my major points. I have increased my score to reflect this.
>
> Regarding your comment "An important aspect of our method is the ability to use a subset of time series in a training batch for model optimization, which improves scalability", this is true but I think this is not a contribution of this paper as this was also present in [1].

---

> > ### Author Response · Authors · 2024-08-12
> >
> > Thank you for your positive feedback and for increasing your score based on our response. We appreciate your acknowledgment of our clarifications.
> >
> > Regarding the use of a subset of time series for model optimization, we agree that this approach was also present in [1]. Our intent was to highlight that our method can also inherit this feature. We will revise the text to ensure this distinction is clear. Thank you for pointing this out.

---

### Official Review · Reviewer_Vi7g · 2024-06-25

**Soundness:** 3
**Presentation:** 3
**Contribution:** 3
**Rating:** 8
**Confidence:** 4

**Summary:**

The paper proposes an approach for multivariate time series forecasting, modelling the correlation between the multivariate errors in close time steps.
The authors show that indeed multivariate errors in close time instants are correlated.
They generalise the work of Zheng et al (Aistats 2024) where the idea of modelling the correlation between errors in different time step has been used only for probabilistic univariate models.
Experimentally, they consider two different deep  architectures suitable for multivariate time series forecasting.
In both cases they report better performance compared to the traditional approach which does not model the correlation between the multivariate errors in different time steps.

**Strengths:**

Generalizing  the model of the temporal correlation of error from the univariate to the multivariate case is a challenging problem.
The proposed approach, based on modelling the correlation of the error as the sum of kernels with different length scales,  can be applied to any deep network model suitable for multivariate forecasting.
The approach is tested with two different neural architectures and positive results are reported in both cases.
The paper is well written.

**Weaknesses:**

None in particular.

**Questions:**

* You show a better CRPS for the models trained with correlation compared to those without correlation. Could you elaborate if the improvement of CRPS is due to better point forecasts, better variance of the predictive distribution, or both?

* line 169-171: could you clarify (also in the final paper) the sentence: Since the parameters of mapping functions are shared for all time series, we can view z_t as a Gaussian process assessed at points hi_t?

**Limitations:**

None in particular.

---

> ### Author Rebuttal · Authors · 2024-08-05
>
> We sincerely appreciate the reviewer's encouraging comments and insightful suggestions.
> # Questions
> > Could you elaborate if the improvement of CRPS is due to better point forecasts, better variance of the predictive distribution, or both?
>
> ***Response:*** In this paper, we use $\operatorname{CRPS}\_{\text{sum}}$ as the main metric, while energy score, quantile loss, and RRMSE are included as additional metrics in the Appendix. The $\operatorname{CRPS}\_{\text{sum}}$ and energy score primarily assess the **overall quality** of the predictive distribution. While quantile loss evaluates the accuracy of specific quantiles, denoted by $\rho$, from the predictive distribution. Specifically, we assess the $0.5$-risk and $0.9$-risk, with $0.5$-risk being equivalent to the MAE of the mean prediction, which reflects the quality of point forecasts. The $0.9$-risk offers a snapshot of the quality of a specific quantile, which is influenced by the variance/covariance of the predictive distribution. The comparison of $0.5$-risk and $0.9$-risk using GPVar as an example is shown in Table 1. We observe that the results vary across different datasets. In some cases, both $0.5$-risk and $0.9$-risk are improved. In other instances, our method improves either the point forecasts or the higher quantile risk. The outcomes can depend on the characteristics (e.g., noise level) of each model and dataset. For GPVar, our method improves $0.5$-risk on 5 out of 9 datasets and $0.9$-risk on 6 out of 9 datasets. **It is important to note that $0.5$-risk and $0.9$-risk provide only two snapshots of the quality of the predictive distribution and are thus not as comprehensive as $\operatorname{CRPS}\_{\text{sum}}$**.
>
> **Table 1. Comparison of $0.5$-risk and $0.9$-risk using GPVar. "w/o" denotes methods without time-dependent errors, while "w/" indicates our method. Mean and standard deviation are obtained from 10 runs of each model.**
> |                           |               | $0.5$-risk    |                       |                        | $0.9$-risk               |                       |
> |---------------------------|---------------|---------------|-----------------------|------------------------|--------------------------|-----------------------|
> |                           | w/o           | w/            | rel. impr.       | w/o                    | w/                       | rel. impr.       |
> | $\mathtt{exchange\\_rate}$ | 0.0109±0.0003 | 0.0095±0.0004 | 12.84% | 0.0042±0.0001 | 0.0057±0.0001 | -35.71% |
> | $\mathtt{solar}$          | 0.4998±0.0025 | 0.5246±0.0016 | -4.96% | 0.1617±0.0004 | 0.1597±0.0003 | 1.24%   |
> | $\mathtt{electricity}$    | 0.0405±0.0003 | 0.0397±0.0002 | 1.98%  | 0.0211±0.0004 | 0.0185±0.0002 | 12.32%  |
> | $\mathtt{traffic}$        | 0.0933±0.0001 | 0.0859±0.0001 | 7.93%  | 0.0666±0.0001 | 0.0580±0.0001 | 12.91%  |
> | $\mathtt{wikipedia}$      | 0.2231±0.0005 | 0.2236±0.0006 | -0.22% | 0.2136±0.0002 | 0.2048±0.0001 | 4.12%   |
> | $\mathtt{m4\\_hourly}$     | 0.0807±0.0001 | 0.0849±0.0001 | -5.20% | 0.0452±0.0002 | 0.0463±0.0001 | -2.43%  |
> | $\mathtt{m1\\_quarterly}$  | 0.2196±0.0023 | 0.1948±0.0005 | 11.29% | 0.3049±0.0044 | 0.2787±0.0027 | 8.59%   |
> | $\mathtt{pems03}$         | 0.0568±0.0000 | 0.0574±0.0001 | -1.06% | 0.0317±0.0000 | 0.0317±0.0001 | 0.00%   |
> | $\mathtt{uber\\_hourly}$   | 0.1035±0.0002 | 0.1013±0.0002 | 2.13%  | 0.0533±0.0002 | 0.0528±0.0001 | 0.94%   |
>
> > line 169-171: could you clarify (also in the final paper) the sentence: Since the parameters of mapping functions are shared for all time series, we can view z_t as a Gaussian process assessed at points hi_t?
>
> ***Response:*** We apologize for any confusion caused. In this paper, the time series variables $\mathbf{z}\_{t}$ jointly follow a multivariate Gaussian distribution. Since any subset of $\mathbf{z}\_{t}$ will also follow a multivariate Gaussian distribution, we can use a training batch comprising $B$ random time series from a total of $N$ time series, on which the likelihood is computed.
>
> Following [1], we assume $\left.\mathbf{z}\_{t} \mid \mathbf{h}\_{t}\right.\sim\mathcal{N}\left( \boldsymbol{\mu}(\mathbf{h}\_{t}), \boldsymbol{\Sigma}(\mathbf{h}\_{t})\right)$, where $\boldsymbol{\Sigma}\_t=\boldsymbol{L}\_{t}\boldsymbol{L}\_{t}^\top+\text{diag}{(\mathbf{d}\_{t})}$, and $\boldsymbol{\mu}\_{t}$, $\boldsymbol{L}\_{t}$, and $\mathbf{d}\_{t}$ are conditioned on $\mathbf{h}\_{t}$:
>
> $$
> \boldsymbol{\Sigma}\left(\mathbf{h}\_t\right)=\left[\begin{array}{ccc}d_1\left(\mathbf{h}\_{1, t}\right) & & 0 \\\ & \ddots & \\\ 0 & & d_N\left(\mathbf{h}\_{N, t}\right)\end{array}\right]+\left[\begin{array}{c}\mathbf{l}_1\left(\mathbf{h}\_{1, t}\right) \\\ \cdots \\\ \mathbf{l}_N\left(\mathbf{h}\_{N, t}\right)\end{array}\right]\left[\begin{array}{c}\mathbf{l}_1\left(\mathbf{h}\_{1, t}\right) \\\ \cdots \\\ \mathbf{l}_N\left(\mathbf{h}\_{N, t}\right)\end{array}\right]^\top=\boldsymbol{D}_t+\boldsymbol{L}\_{t} \boldsymbol{L}\_{t}^\top.
> $$
>
> where $\mu_i$, $d_i$, and $\mathbf{l}_{i}$ are the mapping functions used to transform the hidden state $\mathbf{h}\_{i,t}$ of time series $i$ at time $t$ to the corresponding distribution parameters. By shared mapping functions, we mean that we use a set of global mapping functions $\tilde{\mu}(\cdot)$, $\tilde{d}(\cdot)$, and $\tilde{\mathbf{l}}(\cdot)$ for all time series, instead of having a separate function for each time series. The shared mapping functions enable us to take any subset of time series and calculate the mean and covariance.
>
> [1] Salinas, David, et al. High-dimensional multivariate forecasting with low-rank gaussian copula processes. In NeurIPS, 2019.

---

> > ### Comment · Reviewer_Vi7g · 2024-08-12
> >
> > Thank you for your clarification, I keep unchanged my positive evaluation of the paper.

---

> > > ### Author Response · Authors · 2024-08-12
> > >
> > > Thank you for acknowledging our clarifications and your positive evaluation of our paper.

---

### Official Review · Reviewer_RQB7 · 2024-07-12

**Soundness:** 3
**Presentation:** 3
**Contribution:** 4
**Rating:** 6
**Confidence:** 3

**Summary:**

The paper proposes a method to improve multivariate probability forecasting by accounting for potential temporal dependencies of the residuals. The paper introduces a dynamic covariance using a small number of latent temporal processes. The method is evaluated on standard dataset of multivariate time-series on two forecasting models, with the CRPS as a metric.

**Strengths:**

The method proposed in the paper is original and clearly motivated.
The paper shows improvements of the CRPS accuracy over a number of time-series for two models.

**Weaknesses:**

The paper partly relies on the premise that incorporating time-dependency in the residuals is crucial for providing better uncertainty estimates. Demonstrating that the new residuals generated by the paper's method are less auto-correlated than those without the method is therefore crucial. However, the paper lacks a comprehensive metric for this evaluation. The theoretical introduction could be enhanced for mathematical clarity and efficiency.

**Questions:**

How does the number of time-series influence your method? In particular, it could be interesting to measure the decorrelation of the residuals with your method (with an appropriate metric that averages over all the time-series), with methods without accounting for it.

Minor comments.
- l18 "vital", this is a strong word
- equation (1) is not accurate. Indeed, to simplify, let's take $P=1, Q=2$, and $x = 0$ (in order to remove it). To simplify further, let's assume $z$ takes discrete values. Writing $A=\{z_T=z^0_T\}, B=\{z_{T+1}=z^0_{T+1}\}, C=\{z_{T+2}=z^0_{T+2}\}$, equation (1) implies the equation $\mathbb{P}(B\cap C | A) = \mathbb{P}(B|A)\mathbb{P}(C|B)$ where $\mathbb{P}$ is the probability measure. This identity is wrong. Indeed, one just need to take $A\cap B\cap C = \emptyset$ but $A\cap B \ne \emptyset$ and $B\cap C \ne \emptyset$.

**Limitations:**

The limitations of the method should be explored.

---

> ### Author Rebuttal · Authors · 2024-08-05
>
> We sincerely appreciate the reviewer's encouraging comments and insightful suggestions.
> # Weaknesses
> > The paper partly relies on the premise that incorporating time-dependency in the residuals is crucial for providing better uncertainty estimates. Demonstrating that the new residuals generated by the paper's method are less auto-correlated than those without the method is therefore crucial. However, the paper lacks a comprehensive metric for this evaluation. The theoretical introduction could be enhanced for mathematical clarity and efficiency.
>
> ***Response:*** Thank you for the suggestions. To the best of our knowledge, there is no widely used metric that can comprehensively evaluate both autocorrelation and cross-lag correlation in our case. Therefore, we chose to use the comparison of ACF plots (Figures 6-12 in the Appendix) to demonstrate the effect of decreased autocorrelation and the comparison of cross-correlation plots (Figures 13-18 in the Appendix) to show the effect of decreased cross-lag correlation. We will enhance the theoretical introduction in the next version of the paper to improve the flow.
>
> # Questions
> > How does the number of time-series influence your method? In particular, it could be interesting to measure the decorrelation of the residuals with your method (with an appropriate metric that averages over all the time-series), with methods without accounting for it.
>
> ***Response:*** The number of time series does not influence the training since the model is trained based on a random subset of $B$ time series at a time, regardless of the total number of time series $N$. However, during inference, one can increase the number of time series in a batch to leverage the information from more time series, as long as memory allows. We have provided an additional experiment showing the effect of increasing the number of time series in a batch when performing inference (see Fig. 1 in the PDF for the global rebuttal).
>
> To the best of our knowledge, there is no widely used metric that can comprehensively evaluate both autocorrelation and cross-lag correlation in our case. Therefore, we chose to use the comparison of ACF plots (Figures 6-12 in the Appendix) to demonstrate the effect of decreased autocorrelation and the comparison of cross-correlation plots (Figures 13-18 in the Appendix) to show the effect of decreased cross-lag correlation.
>
> > l18 "vital", this is a strong word.
>
> ***Response:*** Thank you for pointing this out. We will revise the narrative in the next version of the paper.
>
> > equation (1) is not accurate.
>
> ***Response:*** Thank you for your observation. The example given by the reviewer is indeed correct in a general context. However, in Eq.(1), the factorization is based on the joint distribution of Gaussian variables. Since the PDF of a Gaussian is always positive, the factorization in Eq.(1) holds true in this context.
>
> # Limitations
> The limitations of the method should be explored.
>
> ***Response:*** Methodology-wise, the limitations of our method are mainly two-fold. Firstly, there is a potential model misspecification issue by assuming Gaussian noise. A potential solution is to first transform the original observations $\mathbf{z}\_{t}$ into Gaussian-distributed data $\mathbf{x}\_{t}$ using a Gaussian Copula, and then apply our method. The second limitation comes from the parameterization of the covariance matrix. Specifically, the Kronecker structure $\boldsymbol{C}\_t \otimes \mathbf{I}\_{R}$ for the covariance matrix of the latent variable $\boldsymbol{r}\_t^{\text{bat}}$ may be too restrictive. This structure assumes that the rows (i.e., latent temporal processes) in the matrix $\left[\mathbf{r}\_{t-D+1},\ldots,\mathbf{r}\_{t}\right]$ are identically distributed following $\mathcal{N}\left(\boldsymbol{0},\boldsymbol{C}\_t\right)$. Additionally, the parameterization of $\boldsymbol{C}\_t$ could be expanded. Instead of using SE kernels, $\boldsymbol{C}\_t$ could be parameterized as fully learnable positive definite symmetric Toeplitz matrices, allowing for the modeling of negative correlations. Computation-wise, our method can lead to increased training costs due to the larger $DB \times DB$ covariance matrix used in the likelihood calculation.

---

> > ### Comment · Reviewer_RQB7 · 2024-08-12
> >
> > Thank you for the clarifications and the additional experiments.
> > I will maintain my position of accepting this paper.

---

> > > ### Author Response · Authors · 2024-08-12
> > >
> > > Thank you for acknowledging our clarifications and additional experiments.

---

### Official Review · Reviewer_uzY6 · 2024-07-15

**Soundness:** 4
**Presentation:** 4
**Contribution:** 4
**Rating:** 7
**Confidence:** 4

**Summary:**

This paper introduces a method for multi-variate time series forecasting where the residual errors are not assumed to be independent and modeled to be correlated using a Gaussian process model. Standard time-series approaches assume that the errors are temporally independent, however, this assumption does not hold in many datasets due to model misspecification. It is shown experimentally that the residuals have a non-zero correlation between different time steps.

The proposed methods accounts for the correlation between the time steps using a Gaussian process approach. The correlation matrix of the GP is designed to be both temporal and spatial. Spatial correlations occur due to a low rank assumption of the correlation matrix in the spatial dimension for a particular time step. Temporal correlations occur due to a non-identical correlation matrix for the latent variables across different time steps.

Experimental results on benchmark datasets show improved performance with temporal correlations than without. An extensive analysis and interpretation of the results are provided in the appendix.

**Strengths:**

- The presented approach is well motivated as experimentally shown in the extensive analysis. Residual errors are indeed correlated and hence this is an interesting and significant problem in the time series community.
- The proposed method is technically sound. Gaussian processes are an ideal approach to learn correlations between dimensions and hence a natural choice to account for the temporal correlations.
- Experimental results show improved performance on standard benchmark datasets when modeling the correlations. A full set of experiments and extensive analysis shows potential of impact.

**Weaknesses:**

- The proposed method may not be a completely novel approach to time series forecasting. Gaussian processes have been used for time series predictions extensively and a simple way to account for temporal correlations is the following:
  - Learn a deterministic base model on the data, compute the residual and fit a Gaussian process on the residuals.
  - The Gaussian process kernel is a product of two kernels, one along the time dimension and the other across time series.
  - It is unclear how this simple baseline will perform compared to the proposed method which is complex.
- Some strong baselines are missing from the experiments. Just to give some examples:
  - Rangapuram, Syama Sundar, et al. "Deep state space models for time series forecasting." Advances in neural information processing systems 31 (2018).
  - Salinas, David, et al. "DeepAR: Probabilistic forecasting with autoregressive recurrent networks." International journal of forecasting 36.3 (2020): 1181-1191

**Questions:**

- How is the model actually learned? A full algorithm listing all the steps would be very useful.
- Are the base models and residual GP model trained independently? Or are they trained end-to-end?
- How is inference actually done? Let's say we have historical data up to step t, and would like to predict step t+1. The correlation between the residual errors and information about the residual errors till step t helps predict a more accurate residual for step t. Decreasing correlation over longer time ranges means that the advantage of modeling correlated errors will be small for long horizon predictions. Why don't we observe this phenomena in Figures 19 and 20?

**Limitations:**

Please comment on the usefulness of the method for various time ranges. It seems that the method may not be an advantage when predicting over longer time ranges.

---

> ### Author Rebuttal · Authors · 2024-08-05
>
> We sincerely appreciate the reviewer's encouraging comments and insightful suggestions.
>
> # Weaknesses
>
> > The proposed method may not be a completely novel approach to time series forecasting.
>
> ***Response:*** Thank you for your valuable suggestions. Our model is not a two-stage model that first fits the mean of the data and then fits the residuals. Instead, our method employs an end-to-end learning approach that dynamically outputs the hyperparameters of the Gaussian Process (GP) on the errors. This approach avoids the need for constant re-training of the GP to obtain hyperparameters over the temporal horizon in the context of a two-stage model.
>
> > Some strong baselines are missing from the experiments.
>
> ***Response:*** Thank you for your feedback. We did not use Deep SSM and DeepAR as baselines because they are univariate models that do not account for inter-series relationships. Therefore, we included only multivariate baselines in this paper, following [1].
>
> [1] Salinas, David, et al. High-dimensional multivariate forecasting with low-rank gaussian copula processes. In NeurIPS, 2019.
>
> # Questions
> > How is the model actually learned?
>
> ***Response:*** Thank you for your comments. Our model is trained end-to-end, similar to any deep learning model. Therefore, we did not provide a list of algorithm steps, as it is not a multi-stage training process. The base forecasting model provides a hidden state $\mathbf{h}\_{t}$ at each time step. In the baseline models, $\mathbf{h}\_{t}$ is projected onto the distribution parameters $(\boldsymbol{\mu}\_{t}, \boldsymbol{L}\_{t}, \mathbf{d}\_{t})$ using a simple neural network. With our method, $\mathbf{h}\_{t}$ is also projected onto the weight parameters $w\_{m,t}$ of the sum of kernels in the dynamic correlation matrix $\boldsymbol{C}\_t$ using an additional simple neural network. The training process remains the same, with the addition of trainable parameters in the neural networks for projecting $w\_{m,t}$ and a modified loss function (Eq. (10)) that accounts for temporal correlation.
>
> > Are the base models and residual GP model trained independently? Or are they trained end-to-end?
>
> ***Response:*** As clarified in our response to the previous question, our model is trained in an end-to-end manner.
>
> > How is inference actually done?
>
> ***Response:***
> The inference process is performed by applying Eq.(12) **recursively** until the desired prediction range is reached. For example, when predicting $\mathbf{z}\_{t+1}=\boldsymbol{\mu}\_{t+1}+\boldsymbol{\eta}\_{t+1}$:
>
> $$
> \boldsymbol{\eta}\_{t+1} \mid \boldsymbol{\eta}\_{t}, \boldsymbol{\eta}\_{t-1},\ldots,\boldsymbol{\eta}\_{t-D+2}  \sim \mathcal{N} ( \boldsymbol{\Sigma}\_{\star}\boldsymbol{\Sigma}\_{\text{obs}}^{-1}\boldsymbol{\eta}\_{\text{obs}}, \boldsymbol{\Sigma}\_{t+1}- \boldsymbol{\Sigma}\_{\star}\boldsymbol{\Sigma}\_{\text{obs}}^{-1} \boldsymbol{\Sigma}\_{\star}^{\top})
> $$
>
> where $\boldsymbol{\eta}\_{\text{obs}}=\operatorname{vec}\left(\left[\boldsymbol{\eta}\_{t-D+2},\ldots,\boldsymbol{\eta}\_{t-1},\boldsymbol{\eta}\_t\right]\right)$ represents the observed residuals, accessible at forecasting step $t+1$. $\boldsymbol{\Sigma}\_{\text{obs}}$ is the partition of $\boldsymbol{\Sigma}\_{t+1}^{\text{bat}}$ that gives the covariance of $\boldsymbol{\eta}\_{\text{obs}}$, and $\boldsymbol{\Sigma}\_{\star}$ is the partition of $\boldsymbol{\Sigma}\_{t+1}^{\text{bat}}$ that gives the covariance of $\boldsymbol{\eta}\_{t+1}$ and $\boldsymbol{\eta}\_{\text{obs}}$.
>
> **By "recursively", we mean that the samples of $\boldsymbol{\eta}\_{t+1}$ can be used as observed residuals for the next prediction step.** Therefore, the decreasing correlation indicated by $\boldsymbol{C}\_t$ does not imply that as we move further from time $t+1$, $\boldsymbol{\eta}_{\text{obs}}$ becomes less useful. On the contrary, $\boldsymbol{\eta}\_{\text{obs}}$ will be continuously updated as we progress. Thus, it is not necessarily the case that the advantage of modeling correlated errors will diminish for long-horizon predictions.
>
> However, the advantage of modeling error correlation can indeed vary over long-term forecasting. Generally, errors will accumulate and propagate to future time steps when performing autoregressive predictions. Using residuals from previous time steps can be advantageous, especially for non-stationary segments of the time series. However, this may not be the case for time series with strong periodic effects, where the model can leverage information from the seasonal lags of the data. As shown in Figures 19 and 20, the advantage of modeling error correlation decreases over time for some datasets, while for others it does not.
>
> # Limitations
> > Please comment on the usefulness of the method for various time ranges.
>
> ***Response:*** As clarified in an earlier response, it is not necessarily the case that the advantage of modeling correlated errors will diminish for long-horizon predictions. The usefulness of our method in longer-range forecasting can vary based on the quality of predictions generated during the inference process. Generally, errors will accumulate and propagate to future time steps when performing autoregressive predictions. Using residuals from previous time steps can be advantageous, especially for non-stationary segments of the time series. However, this may not be the case for time series with strong periodic effects, where the model can leverage information from the seasonal lags of the data. In cases where the model provides good predictions for long-term forecasting, the advantage of our method can be less obvious.

---

> > ### Comment · Reviewer_uzY6 · 2024-08-10
> >
> > Thanks for the clarifications. I will maintain my position of accepting this paper. I urge the authors to address the above questions fully in the paper. This will only lead to a further improvement in the presentation of the results.
> > - A complete figure showing all the components (and perhaps the loss functions used) will be very helpful for future readers. It helps improve the presentation of the paper.
> > - Limitations: It will be very interesting to see how the error correlations behave over longer horizon forecasts. Both in the case of periodic effects and otherwise.

---

> > > ### Author Response · Authors · 2024-08-12
> > >
> > > Thank you for your constructive feedback. We will add a complete figure showing all components, including the loss functions, to improve the clarity of our paper. We also recognize the value of exploring error correlations over longer forecast horizons and will add a discussion to address this point in a future version of our paper.

---

### Author Rebuttal · Authors · 2024-08-06

Dear AC and Reviwers,

We would like to express our gratitude for your thorough and insightful reviews. We have carefully considered each of your comments and suggestions. Below, we provide a summary of our responses to the main points raised by the reviewers.

1. Clarifications and Misunderstandings
    - Learning and inference process: We have clarified that our model is trained end-to-end and performs inference in an autoregressive manner.
    - Clarification of viewing $\mathbf{z}_{t}$ as a Gaussian process.
2. Comparison with Related Work
    - Is the method competitive with better likelihood? We have added experiments showing that our method is competitive with a likelihood based on the $t$-distribution (Table 1 in pdf).
3. Experimental Results
    - How does the number of time series influence our method? We have added experiments showing the effect of increasing the number of time series in a batch when performing inference (Figure 1 in pdf).
    - How can the decorrelation of the residuals be measured? We have shown the decorrelation effect in our Appendix using ACF plots and cross-correlation plots.
    - Does the improvement in CRPS come from the point forecast or better variance/covariance? We have provided quantile loss as two snapshots from the predictive distribution to assess the improvement (Table 1 in the response to Reviewer Vi7g).
    - Experiment details: We have provided information about the number of parameters in each component of our models (Table 2&3 in pdf).

We believe that these revisions have significantly strengthened the paper and addressed the reviewers' concerns. We appreciate the reviewers’ time and effort in providing constructive feedback, which has been invaluable in improving our work.

Thank you for your consideration.

Sincerely,

The authors

---

### Decision · Program_Chairs · 2024-09-25

**Decision:**

Accept (poster)

**Comment:**

The reviews are generally strong and agreement is high. I would strongly
encourage the authors to incorporate the feedback and suggestions into their
revision, as well as their clarifications. Please prioritize the addition of a figure
showing all components the discussion of error correlations over longer forecast
horizons. Please also be sure to modify the paper along the lines of the author
rebuttal, especially to reviewer szpP.